# PAID: Pairwise Angular-Invariant Decomposition for Continual Test-Time Adaptation

**Kunyu Wang, Xueyang Fu, Yuanfei Bao, Chengjie Ge, Chengzhi Cao,
Wei Zhai, Zheng-Jun Zha**[*]
University of Science and Technology of China
{kunyuwang, baoyuanfei, cjge, chengzhicao}@mail.ustc.edu.cn
{xyfu, wzhai056, zhazj}@ustc.edu.cn

## Abstract

Continual Test-Time Adaptation (CTTA) aims to online adapt a pre-trained model to changing environments during inference. Most existing methods focus on exploiting target data, while overlooking another crucial source of information, the pre-trained weights, which encode underutilized domain-invariant priors. This paper takes the geometric attributes of pre-trained weights as a starting point, systematically analyzing three key components: magnitude, absolute angle, and pairwise angular structure. We find that the pairwise angular structure remains stable across diverse corrupted domains and encodes domain-invariant semantic information, suggesting it should be preserved during adaptation. Based on this insight, we propose **PAID** (**P**airwise **A**ngular-**I**nvariant **D**ecomposition), a prior-driven CTTA method that decomposes weight into magnitude and direction, and introduces a learnable orthogonal matrix via Householder reflections to globally rotate direction while preserving the pairwise angular structure. During adaptation, only the magnitudes and the orthogonal matrices are updated. PAID achieves consistent improvements over recent SOTA methods on four widely used CTTA benchmarks, demonstrating that preserving pairwise angular structure offers a simple yet effective principle for CTTA. Our code is available at `https://github.com/wangkunyu241/PAID`.

## 1 Introduction

Deep Neural Networks (DNNs) have achieved remarkable success in various computer vision tasks [37, 38, 71–73, 80]. However, when there exist domain discrepancies between training and testing environments [14, 47, 57, 58], directly applying a source pre-trained model may cause significant performance degradation, particularly when the target distribution is unpredictable and continually changing over time. This challenge in real-world scenarios has motivated the emergence of Continual Test-Time Adaptation (CTTA) [60], which aims to adapt source pre-trained models during inference to evolving test data, making it especially suitable for practical applications.

In CTTA, models rely on two key sources of information: the prior knowledge encoded in pre-trained source weights and the streaming data from the target domains. Most existing methods [26, 62, 63] focus on adapting to the target data, while treating pre-trained weights as static initialization, with their potential largely overlooked. However, these weights, learned from large-scale supervised training [4, 48], may encode transferable priors that remain invariant across domains. We posit that leveraging such invariances in parameter space can help address core challenges in CTTA, including catastrophic forgetting and error accumulation. This work explores this direction by uncovering

---

[*]Corresponding Author

39th Conference on Neural Information Processing Systems (NeurIPS 2025).

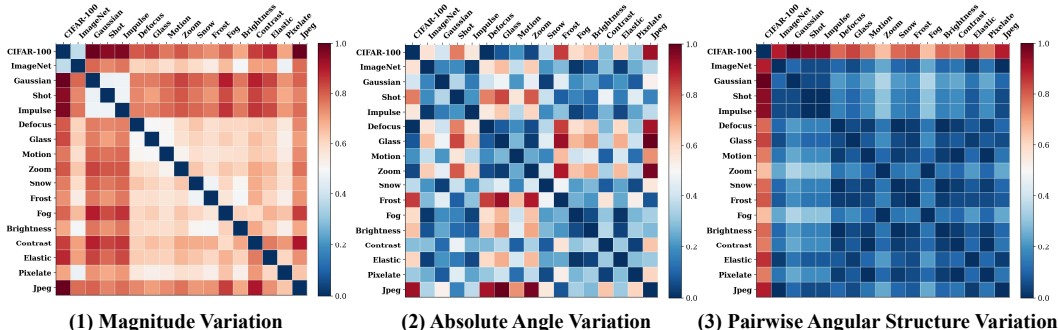

| (1) Magnitude Variation | (2) Absolute Angle Variation | (3) Pairwise Angular Structure Variation |

Figure 1: (**Experiment 1**) Visualization of cross-domain variation of three geometric properties. Pairwise angular structure remains stable under corruption but varies under semantic shift, suggesting it encodes semantic-relevant, domain-invariant information. In contrast, magnitude and absolute angle fluctuate irregularly across domains, reflecting domain-specific shifts.

domain-invariant components in pre-trained weights, offering a new perspective on knowledge retention and transfer for continual adaptation.

## 1.1 Motivation

Recent studies in hyperspherical learning [31–34, 52] reveal that the angular component of neuron weights, rather than their magnitudes, primarily encodes discriminative semantic information crucial for visual recognition. Similar observations have been made in generative models [46], where the pairwise angular structure among neurons effectively preserve semantic consistency after fine-tuning on different domains. These findings suggest that beyond treating neural weights as individual scalars, the geometry of weight space, particularly the angular components, may encode invariant semantic priors derived from pre-training. Motivated by this, we ask: can the angular configuration of pre-trained weights serve as a stable semantic anchor during continual adaptation? To investigate this hypothesis, we design three sets of targeted experiments that isolate and examine the respective roles of magnitude and angular components in the adaptation process. [2]

Specifically, neuron weights, as exemplified by the linear projection matrix $W = [w_1, w_2, \ldots, w_k] \in \mathbb{R}^{d \times k}$, can be decomposed into a magnitude $M \in \mathbb{R}^{1 \times k}$ and a unit-length direction $\hat{W} \in \mathbb{R}^{d \times k}$ :

$$W = M \odot \hat{W}, \quad \text{where} \quad w_i = \|w_i\| \cdot \hat{w}_i, \quad \hat{w}_i = \frac{w_i}{\|w_i\|}, \quad \|\hat{w}_i\| = 1. \tag{1}$$

Building on this view, we identify three geometric attributes of the weight space that are relevant to adaptation: (1) the magnitude, which determines the scaling of feature responses; (2) the absolute angle, referring to the orientation of each unit direction vector in the feature space, which may rotate during adaptation; (3) the pairwise angular structure, defined as the set of angles between all pairs of unit direction vectors, capturing how weight vectors are arranged relative to one another.

**Experiment 1** examines the domain invariance of geometric attributes in weight space by analyzing their variation under different conditions. We use a ViT-Base model pre-trained on ImageNet and conduct two types of experiments: (1) Corruption, where the model performs test-time adaptation on each of the 15 corrupted domains in ImageNet-C; and (2) Semantic shift, where the model is fine-tuned on CIFAR-100 to induce semantic changes, serving as a reference. To quantify the average variation in geometric attributes, we define three metrics: (1) magnitude variation $\Delta M$, (2) absolute angle variation $\Delta A$, and (3) pairwise angular structure variation $\Delta S$, quantified using hyperspherical energy [31, 46], which refers to the sum of hyperspherical similarity between all pairwise neurons:

$$\Delta M(W^1, W^2) = \frac{1}{k} \sum_{n=1}^{k} \left| \|w_n^1\| - \|w_n^2\| \right|, \quad \Delta A(W^1, W^2) = \frac{1}{k} \sum_{n=1}^{k} \left( 1 - \cos(\hat{w}_n^1, \hat{w}_n^2) \right), \tag{2}$$

$$\Delta S(W^1, W^2) = \left| \sum_{i \neq j} \left\| \hat{w}_i^1 - \hat{w}_j^1 \right\|^{-1} - \sum_{i \neq j} \left\| \hat{w}_i^2 - \hat{w}_j^2 \right\|^{-1} \right|, \quad \text{where } i, j \in \{1, \ldots, k\}, \tag{3}$$

---

[2]More experimental details can be found in Appendix A

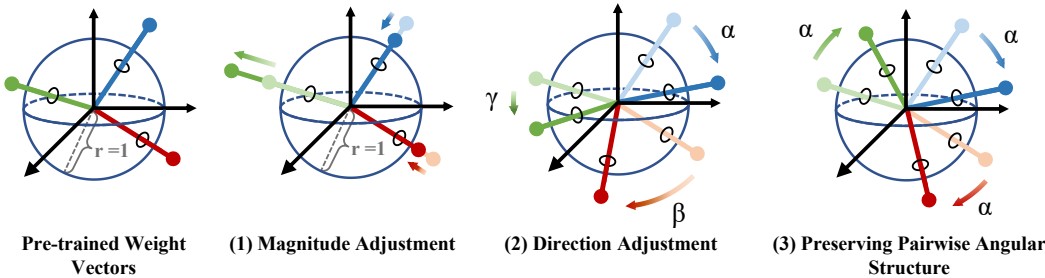

Pre-trained Weight Vectors    (1) Magnitude Adjustment    (2) Direction Adjustment    (3) Preserving Pairwise Angular Structure

Figure 2: (**Experiment 2**) Illustration of three update strategies. (1) Scaling the magnitude of each vector; (2) Independently rotating each vector, altering their absolute angle; (3) Jointly rotating all vectors while preserving their pairwise angular structure.

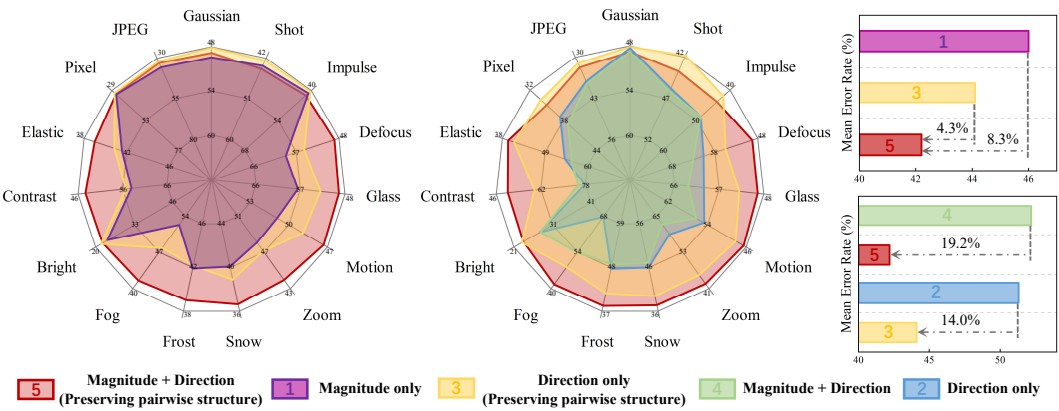

Figure 3: (**Experiment 2**) Radar and bar charts showing classification error rates across 15 corruption domains and their mean. Comparisons between settings (5 vs 1, 3; 5 vs 4; 3 vs 2) show that pairwise angular structure is a domain-invariant component worth preserving, while magnitude and direction, when constrained by fixed angular structure, are domain-specific and beneficial for adaptation.

As shown in Fig. 1, the pairwise angular structure remains consistently stable across all corruption domains, with minimal changes in hyperspherical energy. In contrast, it exhibits substantial variation under semantic shift. This contrast reveals that pairwise angular structure is well preserved under perturbations and varies in response to semantic changes. In contrast, both magnitude and absolute angle exhibit irregular variations across domains, indicating that their changes are driven by adaptation to domain-specific statistical differences rather than semantics. These findings provide strong empirical support that pairwise angular structure encodes semantic-relevant, domain-invariant properties, whereas magnitude and absolute direction are more responsive to domain-specific statistical shifts.

**Experiment 2** conducts an ablation study to clarify the contribution of geometric attributes to adaptation. Based on the decomposition, we design three update strategies, as shown in Fig. 2: (1) magnitude adjustment, enabling learning of the magnitude matrix $M$; (2) direction adjustment, enabling learning of unit direction matrix $\hat{W}$; and (3) direction adjustment while preserving pairwise angular structure, implemented via orthogonal rotation [5, 16, 69] of direction matrix. Note that changing the pairwise angular structure inherently changes absolute angle, but not vice versa. Thus, (3) represents a constrained subset of (2). Using these strategies, we construct five settings for continually adapting a ViT-Base model (pre-trained on ImageNet) to the 15 ImageNet-C domains: (1) magnitude only; (2) direction only; (3) direction only while preserving pairwise structure; (4) magnitude + direction; and (5) magnitude + direction while preserving pairwise structure.

As shown in Fig. 3, setting (5) outperforms settings (1) and (3), while settings (2) and (4), both of which alter the pairwise angular structure, lead to a clear performance drop compared to (3) and (5). This contrast underscores the role of pairwise angular structure as a domain-invariant component that should be preserved during adaptation. In contrast, magnitude and direction, when adjusted under the constraint of maintaining relative angular geometry, serve as domain-specific components that are adaptable and beneficial for enhancing performance in CTTA.

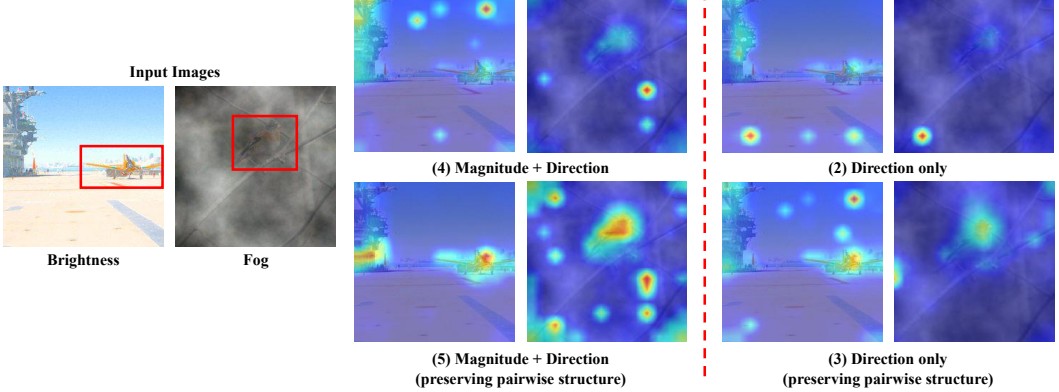

Figure 4: (**Experiment 3**) Attention map visualizations for two setting pairs (5 vs 4; 3 vs 2). The comparison further supports the semantic relevance and domain invariance of pairwise angular structure in CTTA.

**Experiment 3** further substantiates the earlier conclusions from a perceptual perspective by visualizing attention maps under the five settings in Experiment 2. As shown in Fig. 4, preserving the pairwise angular structure allows the model to focus on crucial semantic regions. In contrast, settings (2) and (4) that alter this structure lead to diffused and misaligned attention, with the model failing to capture core object information. This comparison reinforces the critical role of preserving pairwise angular structure for achieving cross-domain invariance.

## 1.2 Contribution

These three sets of experiments—statistical analysis, functional validation, and visual interpretation—collectively reveal a key insight: the pairwise angular structure of neural weights encodes a semantically relevant and domain-invariant prior derived from pre-training, which should be preserved during CTTA. In contrast, the magnitude and absolute angle, when adjusted under the constraint of preserving the pairwise angular structure, serve as domain-specific components that enable effective adaptation to the target domains.

Building on this insight, we propose Pairwise Angular-Invariant Decomposition (PAID) for CTTA. PAID explicitly decomposes pre-trained weights into magnitude and direction matrices. To maintain the pairwise angular structure during adaptation, we introduce a learnable orthogonal matrix constructed via Householder transformations, enabling global rotation of directions without altering their relative angular configuration. Leveraging this orthogonality, we freeze the original direction matrix and update only the magnitude and the injected orthogonal matrix, ensuring structure-preserving adaptation. In summary, the contributions can be summarized as follows:

- We identify the pairwise angular structure of pre-trained weights as a domain-invariant semantic prior that should be preserved during CTTA, supported by statistical, functional, and visual analyses.
- We propose PAID, a novel prior-driven CTTA method that preserves the pairwise angular structure of weights while enabling controlled adaptation of their magnitudes and directions through orthogonal transformations.
- PAID achieves consistent improvements over recent SOTA methods on four standard CTTA benchmarks, demonstrating strong effectiveness and generalizability.

## 2 Related Work

**Continual Test-time Adaptation (CTTA)** aims to online adapt a source pre-trained model to handle a sequence of target domains. Existing learning paradigms can be broadly categorized into three types [26, 56, 62, 63]. Optimization-based methods aim to adjust pre-trained models by designing new objectives, including statistics calibration [11, 54, 59, 68], consistency regularization [29, 51, 53, 66], entropy minimization [41, 55, 78], and pseudo-labeling [1, 20, 70]. Data-based methods focus on

enhancing data diversity or mitigating the impact of distributional shifts. Typical strategies include data augmentation [8, 36, 74] and memory bank [12, 61]. Model-based methods enhance adaptation by modifying or extending the model architecture, including module addition [20, 29], module substitution [19], and prompt-based mechanisms [9, 10, 50, 76]. While similar in form to model-based methods, our method introduces a novel perspective by leveraging transferable priors encoded in pre-trained weights. We exploit the pairwise angular structure as a domain-invariant prior to enhance CTTA.

**Parameter-Efficient Fine-Tuning (PEFT)** [13] reduces adaptation cost by freezing most large model parameters and updating only a small, task-specific subset. Representative methods include adapter-based methods [15], low-rank adaptation [6, 17, 75] and prompt tuning [22, 45]. Among them, DoRA [30] and OFT [46] are particularly insightful: DoRA decomposes weights into magnitude and direction, applying low-rank updates solely to the direction for near full-tuning expressiveness; OFT adopts layer-shared orthogonal transformations to fine-tune only the direction of weights, preserving the angular structure critical for semantic consistency in generative models. Inspired by these, we introduce the idea of weight decomposition into the field of test-time adaptation, rethinking it from the lens of generalization. Through a systematic analysis of pre-trained weight space, we find that the pairwise angular structure remains stable across domains and encodes domain-invariant semantics. This observation, supported by statistical, functional, and visual evidence, motivates us to explicitly preserve angular structure during CTTA.

## 3 Methodology

### 3.1 Preliminaries

Given a model $f_\theta$ pre-trained on the source domain $D_s = \{x_s, y_s\}$, our goal is to adapt this model to a sequence of continually changing target domains $\{D_t^1, D_t^2, \ldots, D_t^N\}$. In an online setting, the model $f_\theta$ processes a sequence of test data batches $\{B_t\}_{t=1}^\infty$, with each batch $B_t$ arriving at time step $t$. Consistent with prior work [60], we assume that all samples in a batch $B_t$ come from the same target domain, though the domain identity is unknown. At each time step $t$, CTTA aims to adapt the model parameters from $\theta_t$ to $\theta_{t+1}$ by learning from the current batch $B_t$, thereby enhancing performance on subsequent batches.

A Vision Transformer model consists of multiple encoder layers, each containing a Multi-Head Attention (MHA) block and a Feed-Forward Network (FFN) block. The MHA block utilizes three key linear weights $W_q$ for the query, $W_k$ for the key, and $W_v$ for the value to compute attention scores and aggregate weighted values from normalized feature representations. In addition, a linear weight $W_o$ combines the outputs of all attention heads. In the FFN block, the input is processed through two linear weights $W_{m_1}$ and $W_{m_2}$ with a GELU function applied between them. This work focuses on decomposing the pre-trained weights of these linear layers $W_q, W_k, W_v, W_o, W_{m_1}$ and $W_{m_2}$, preserving their domain-invariant priors to enhance CTTA task. Fig. 5 provides an overview of our method.

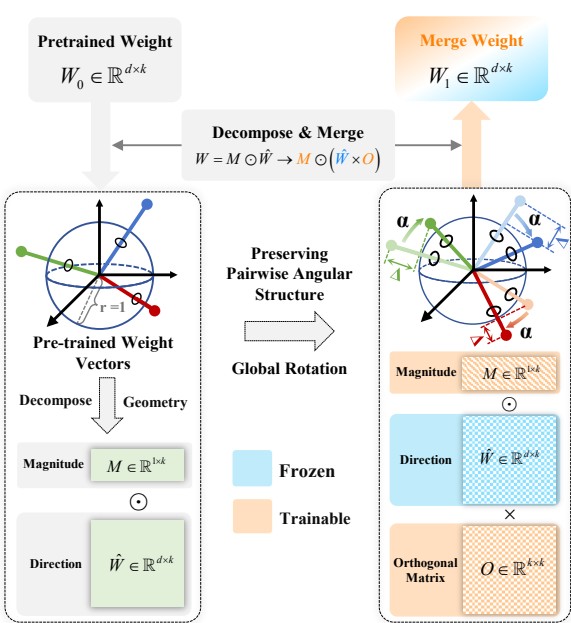

Figure 5: PAID decomposes pre-trained weights into magnitude and direction. To preserve pairwise angular structure, we introduce a learnable orthogonal matrix, enabling global rotation. Only magnitude and orthogonal matrices are updated during adaptation.

## 3.2 Pairwise Angular-Invariant Decomposition (PAID)

Following Eq. 1, a linear projection matrix $W = [w_1, w_2, \ldots, w_k] \in \mathbb{R}^{d \times k}$ can be decomposed into a magnitude matrix $M \in \mathbb{R}^{1 \times k}$ and a direction matrix $\hat{W} = [\hat{w}_1, \hat{w}_2, \ldots, \hat{w}_k] \in \mathbb{R}^{d \times k}$ as:

$$W = M \odot \hat{W}, \quad \text{where} \quad w_i = \|w_i\| \cdot \hat{w}_i, \quad \|\hat{w}_i\| = 1. \tag{4}$$

Empirical evidence suggests that the pairwise angular structure of pre-trained weights, defined as the angles between all pairs of unit direction vectors, encodes a semantically meaningful and domain-invariant prior. This structure should be preserved during the CTTA process. In contrast, magnitude and absolute angle, when adapted under the constraint of preserving this structure, serve as domain-specific components that support effective adaptation. Accordingly, we make the magnitude matrix $M$ learnable while freezing the original direction matrix $\hat{W}$. To enable directional adaptation without disrupting angular structure, we introduce a learnable orthogonal matrix $O \in \mathbb{R}^{k \times k}$, which performs global rotations while preserving pairwise angular structure:

$$M \odot \hat{W} \longrightarrow M \odot \left( \hat{W} \cdot O \right), \tag{5}$$

A real square matrix $O \in \mathbb{R}^{k \times k}$ is orthogonal if it satisfies $O^\top O = I$. Such matrix represent distance-preserving linear transformation, including rotation and reflection. For any vector $x \in \mathbb{R}^k$, the transformation $x \mapsto Ox$ preserves the Euclidean norm, that is, $\|Ox\| = \|x\|$. For any pair of vectors $(x, y)$, it also preserves inner products: $\langle Ox, Oy \rangle = \langle x, y \rangle$. As a result, the relative geometry among the vectors remains unchanged. Therefore, applying an orthogonal transformation to a set of vectors performs a global rotation without altering their pairwise angular structure.

To construct the orthogonal transformation, we adopt the Householder reflection formulation [5, 16, 69]. A Householder reflection is a linear transformation that reflects a vector across a hyperplane perpendicular to a unit vector. Given $u \in \mathbb{R}^k$ with $\|u\|_2 = 1$, the corresponding Householder matrix is defined as:

$$H = I - 2uu^\top, \tag{6}$$

where $H \in \mathbb{R}^{k \times k}$ is orthogonal and symmetric, satisfying $H^\top H = I$ and $\det(H) = -1$, indicating a reflection. The transformation preserves Euclidean norms and inner products, i.e.,

$$\|Hx\|_2 = \|x\|_2, \quad \langle Hx, Hy \rangle = \langle x, y \rangle, \quad \forall x, y \in \mathbb{R}^k, \tag{7}$$

thereby maintaining both vector lengths and pairwise angles. Since Householder matrices are orthogonal, and the product of orthogonal matrices remains orthogonal, a general orthogonal matrix $O \in \mathbb{R}^{k \times k}$ can be constructed as a chain of $r$ Householder reflections:

$$O = \prod_{i=1}^{r} H_i = \prod_{i=1}^{r} (I - 2u_i u_i^\top), \quad u_i \in \mathbb{S}^{k-1}. \tag{8}$$

This parameterization is expressive: when $r = k$, it can represent any element in the orthogonal group $O(k)$, while smaller $r$ yields a trade-off between representational capacity and efficiency. Therefore, we adopt such a chain to construct the learnable orthogonal transformation used for structure-preserving adaptation and $r$ denotes orthogonal matrix coefficient.

## 3.3 Optimization Objective

In line with prior CTTA works [39, 40, 77], we adopt a feature distribution alignment strategy between the source and target domains. The core idea is to reduce the domain shift by aligning the first and second moments (i.e., mean and standard deviation) of features extracted from both domains. To ensure consistency, all features are extracted from the same location in the backbone: the CLS token output after the final layer normalization and before the classification head in ViT-Base. This representation is used for both source and target domains throughout the CTTA process.

We pre-compute the source domain statistics $(\mu_s, \sigma_s)$ using a randomly sampled subset of 500 images from the source domain $D_s$. This process is performed offline, and no further access to source data is needed at test time. The resulting statistics are compact and stored for use during adaptation. At each test-time step $T$, a new target batch $B_t^T$ arrives, where the subscript $t$ indicates the target domain. The features of $B_t^T$, denoted as $Z_t^T$, are extracted from the same CLS token position. We compute

the batch-wise mean $\mu_t^T$ and standard deviation $\sigma_t^T$ of the target features. To align the source and target distributions, we define the following objective:

$$\mathcal{L} = \|\mu_s - \mu_t^T\|_2 + \lambda\|\sigma_s - \sigma_t^T\|_2, \tag{9}$$

where $\lambda$ is a weighting hyper-parameter that balances the contribution of the two terms.

### 3.4 Intuitive Explanation of PAID

To explain the intuition for why PAID works, we draw an analogy from the frequency domain decomposition of images. In the Fourier transform [43], an image can be decomposed into a magnitude and a phase spectrum. The magnitude captures energy distribution, contrast, and intensity, and is highly sensitive to domain-specific variations such as noise and style. In contrast, the phase encodes angular alignment with Fourier bases, which determines the structural layout and semantic content. The separation between semantic structure and domain-specific appearance has been widely explored in frequency-based domain generalization [18, 21, 25, 27, 67]. For instance, APR [2] reveals that networks are more sensitive to magnitude perturbations, while phase is crucial for retaining semantic information and achieving robust recognition. FACT [65] verifies the domain-invariant property of phase by showing that transferring phase yields better generalization than amplitude. This perspective supports our design in PAID. The pairwise angular structure, akin to the phase spectrum, captures semantics and remains invariant across domains. PAID preserves the angular structure while adapting the magnitude and direction via constrained orthogonal transformations, thus achieving structure-preserving adaptation across domains.

### 3.5 Theoretical Justification of PAID

To justify the design of PAID, we analyze the effect of domain shifts on the angular structure of weights using a simplified linear classification model. This reveals why a shared orthogonal transformation suffices for adapting to domain corruptions but not semantic shifts.

**Setup.** Let $x \in \mathbb{R}^a$ be an input vector and $w_c \in \mathbb{R}^a$ the weight vector for class $c \in \{1, \ldots, C\}$. The predicted label is:

$$\hat{y}(x) = \arg\max_c w_c^\top x.$$

**Case 1: Corruption (Class-agnostic Transformation).** Suppose all inputs are transformed by the same differentiable function $x' = d(x)$. Using first-order Taylor expansion around clean data:

$$d(x) \approx Sx + q,$$

where $S \in \mathbb{R}^{a \times a}$ is the Jacobian matrix, assumed invertible and independent of class $y$. Under this assumption, the optimal target-domain classifier that preserves original class boundaries satisfies:

$$w_c' = S^{-\top} w_c, \quad \forall c.$$

We write $S^{-\top} = RH$ (polar decomposition), where $R$ is orthogonal ($R^\top R = I$) and $H$ is symmetric positive definite. The transformed weights are:

$$w_c' = RHw_c.$$

The cosine similarity between any two transformed weights is:

$$\cos\angle(w_c', w_d') = \frac{w_c^\top H^2 w_d}{\|Hw_c\| \, \|Hw_d\|}.$$

**Near-isotropic assumption.** We assume $H^2 \approx \gamma^2 I$ for some scalar $\gamma > 0$. This holds when the degradation has approximately equal effect in all directions, such as Gaussian blur, light noise, or JPEG compression. Under this mild and physically reasonable assumption, pairwise angles are approximately preserved:

$$\cos\angle(w_c', w_d') \approx \cos\angle(w_c, w_d).$$

**Case 2: Semantic Shift (Class-dependent Transformation).** Now suppose each class undergoes a different transformation:

$$x' = d_c(x) \approx S_c x + q_c, \quad S_c \in \mathrm{GL}(a).$$

Table 1: Mean classification error rate (%) and gain (%) on ImageNet-to-ImageNet-C using ViT-Base. **Bold** indicates the best performance. Fine-grained performances are shown in Appendix C.

| Method | Source | Pseudo [24] | TENT [55] | CoTTA [60] | VDP [9] | SAR [42] | RoTTA [68] | EcoTTA [51] | ViDA [29] | C-MAE [28] | Ours |
|--------|--------|-------------|-----------|------------|---------|----------|------------|-------------|-----------|------------|------|
| Mean↓ | 55.8 | 66.8 | 51.0 | 54.8 | 50.0 | 45.6 | 48.2 | 48.0 | 43.4 | 42.5 | **42.2** |
| Gain↑ | 0.0 | -11.0 | +4.8 | +1.0 | +5.8 | +10.2 | +7.6 | +7.8 | +12.4 | +13.3 | **+13.6** |

Table 2: Mean classification error rate (%) and gain (%) on CIFAR100-to-CIFAR100-C using ViT-Base. **Bold** indicates the best performance. Fine-grained performances are shown in Appendix C.

| Method | Source | Pseudo [24] | TENT [55] | CoTTA [60] | VDP [9] | ViDA [29] | C-MAE [28] | Ours |
|--------|--------|-------------|-----------|------------|---------|-----------|------------|------|
| Mean↓ | 35.4 | 33.2 | 32.1 | 34.8 | 32.0 | 27.3 | 26.4 | **24.9** |
| Gain↑ | 0.0 | +2.2 | +3.3 | +0.6 | +3.4 | +8.1 | +9.0 | **+10.5** |

The new classifier becomes:

$$w_c'' = S_c^{-\top} w_c.$$

Since $S_c$ varies across classes, there is generally no single matrix $T$ such that $w_c'' = T w_c$ for all $c$, and therefore:

$$\cos \angle(w_c'', w_d'') \neq \cos \angle(w_c, w_d).$$

Semantic shifts necessarily change the pairwise angular structure.

**Conclusion.** When domain shifts are class-agnostic and locally differentiable, they can be represented by a shared invertible matrix $S$, and the optimal classifier corresponds to applying the inverse transpose $S^{-\top}$ to all weights. This transformation preserves angles up to a near-isotropic assumption, justifying why PAID only needs to learn a shared orthogonal rotation and magnitude adjustment. When the shift is class-dependent, this shared structure no longer exists, and angular geometry must change.

## 4 Experiments

In Section 4.2, we compare our method with state-of-the-art approaches on both classification and segmentation CTTA benchmarks. Section 4.3 analyzes the effectiveness of our design choices and investigates the influence of the injected matrix coefficient, test batch size, number of source examples, and computational overhead. Additional ablations, including the effect of the loss coefficient, the choice of injection layer, and a 10-round classification CTTA evaluation, and experiments on convolutional backbones, are provided in Appendix B and D.

### 4.1 Experimental Setup

**Datasets.** We evaluate our method on three classification CTTA benchmarks: CIFAR10-to-CIFAR10C, CIFAR100-to-CIFAR100C [23], and ImageNet-to-ImageNet-C [14]. In the classification tasks, we follow the sequential adaptation process described in [51], where the pre-trained source model adapts to each of the 15 target domains, each defined by the highest corruption severity. Online prediction results are immediately assessed after processing the input. For segmentation CTTA, we assess our method on Cityscapes-to-ACDC, where Cityscapes [3] serves as the source domain and ACDC [49] as the target domains, which includes images captured under four distinct unobserved visual conditions: Fog, Night, Rain, and Snow. To simulate continual environmental changes, we cyclically iterate through the same sequence of target domains (Fog → Night → Rain → Snow) three rounds, reflecting real-world scenarios.

**Methods Compared.** We compare our method against several strong CTTA baselines, including Source, Pesudo [24], TENT [55], CoTTA [60], DePT [10], VDP [9], SAR [42], RoTTA [68], EcoTTA [51], ViDA [29], and C-MAE [28]. "Source" represents the use of the pre-trained model for adaptation without any specific method. The selection of comparison methods is based on their open-source availability and representativeness, with implementations and results drawn from publicly available codebases, paper descriptions, and established benchmarks.

**Implementation Details.** For the classification CTTA tasks, we use ViT-base [7] as the backbone model, resizing input images to 384×384 for CIFAR10-C and CIFAR100-C, and to 224×224 for the ImageNet-C benchmark. For the segmentation CTTA task, we employ Segformer-B5 [64] pre-trained on the Cityscapes dataset as the source model, down-sampling input images from 1920×1080 to

Table 3: Mean classification error rate (%) and gain (%) on CIFAR10-to-CIFAR10-C using ViT-Base. **Bold** indicates the best performance. Fine-grained performances are shown in Appendix C.

| Method | Source | Pseudo [24] | TENT [55] | CoTTA [60] | VDP [9] | ViDA [29] | C-MAE [28] | Ours |
|---|---|---|---|---|---|---|---|---|
| Mean↓ | 28.1 | 26.9 | 23.5 | 24.6 | 24.1 | 20.7 | 12.6 | **11.0** |
| Gain↑ | 0.0 | +1.2 | +4.6 | +3.5 | +4.0 | +7.4 | +15.5 | **+17.1** |

Table 4: Mean mIoU (%) and gain (%) on Cityscapes-to-ACDC (3-round average) using Segformer-B5. **Bold** indicates the best performance. Fine-grained performances are shown in Appendix C.

| Metric | Source | TENT [55] | CoTTA [60] | DePT [10] | VDP [9] | SAR [42] | EcoTTA [51] | ViDA [29] | C-MAE [28] | Ours |
|---|---|---|---|---|---|---|---|---|---|---|
| Mean↑ | 56.7 | 55.7 | 58.6 | 53.4 | 58.2 | 57.0 | 55.8 | 61.9 | 61.8 | **62.2** |
| Gain↑ | 0.0 | -1.0 | +1.9 | -3.3 | +1.5 | +0.3 | -0.9 | +5.2 | +5.1 | **+5.5** |

$960 \times 540$. The AdamW [35] optimizer is used with parameters $(\beta_1, \beta_2) = (0.9, 0.999)$. Hyper-parameters for CIFAR10-C, CIFAR100-C, ImageNet-C, and ACDC are set as follows: batch size $\{64, 64, 64, 1\}$, orthogonal matrix coefficient $\{12, 12, 12, 12\}$, and loss coefficient $\{1.0, 1.0, 0.1, 1.0\}$. To initialize the learnable parameters, we perform warm-up iterations on classification datasets. For linear layer selection, we inject all linear layers, including those in the multi-head attention block (q, k, v, o) and the MLP block (m). The number of source examples is set to 500. The experiments are conducted on the NVIDIA RTX 3090 GPU. All ablation studies are conducted on ImageNet-to-ImageNet-C unless otherwise specified.

## 4.2 Results on Benchmark Datasets

Table 1 reports classification error rates on ImageNet-C under corruption severity level 5. Our method achieves the lowest mean error of 42.2%, outperforming strong baselines such as SAR (45.6%), ViDA (43.4%), and C-MAE (42.5%). Compared to the source model, our method yields a substantial improvement of +13.6%. In Table 2, we present results on CIFAR100-C. Our method achieves the best overall performance with a mean error of 24.9%, surpassing recent leading methods including ViDA (27.3%) and C-MAE (26.4%). The performance gain over the source model reaches +10.5%. Table 3 shows the results on the simpler CIFAR10-C dataset. Our method achieves a mean error of 11.0%, which is a +17.1% improvement over the source model and also outperforms the best baseline (C-MAE at 12.6%). This confirms the effectiveness of our adaptation mechanism even on small-scale datasets. For the segmentation task, Table 4 reports the average mIoU on the Cityscapes-to-ACDC benchmark over three consecutive adaptation rounds. Our method achieves the highest average mIoU of 62.2%, showing robust performance across rounds and delivering a consistent +5.5% improvement over the source model.

## 4.3 Ablation Analysis

**Effect of Each Component.** We conduct ablation studies to validate the effectiveness of key design choices in PAID, as shown in Tab. 5. A central finding is that preserving the pairwise angular structure during adaptation is essential for strong CTTA performance. Directly adjusting the direction matrix, which allows each weight vector to rotate independently, disrupts the relative geometry and leads to performance degradation. In contrast, PAID freezes the original direction matrix and applies a learnable orthogonal transformation to rotate all vectors jointly. This enables adjustment of absolute angles while preserving their relative angular structure. We also compare two general parameter-efficient tuning methods: LoRA [17], which adds low-rank matrices to linear layers, and DoRA [30], which decomposes linear weights into magnitude and direction and applies LoRA to the directional component. Both methods perform sub-optimally in our setting, suggesting that generalization in CTTA demands tailored, customized designs.

**Effect of Injected Matrix Coefficient.** We perform an ablation study on the number of orthogonal matrices used in the Householder transformation chain (denoted as the coefficient $r$), as shown in Fig. 6 (a). While increasing the number of matrices enhances the model's expressive capacity, we observe that CTTA performance does not improve monotonically with the number of matrices. A possible reason is that the corrections required for domain perturbations do not demand overly complex transformations, and higher complexity may negatively affect model convergence in online adaptation.

Table 5: Ablation on design choices. "Adjust Magn." denotes whether the magnitude matrix is updated, "Adjust Dir." denotes whether the direction matrix is updated, and "Inject Orth." indicates whether the direction matrix is frozen and rotated via orthogonal matrices.

| Adjust Magn.? | Adjust Dir.? | Inject Orth.? | Mean↓ | Gain↑ |
|---|---|---|---|---|
| Baseline | | | 55.8 | 0.0 |
| ✓ | ✗ | ✗ | 46.0 | +9.8 |
| ✗ | ✓ | ✗ | 51.3 | +4.5 |
| ✓ | ✓ | ✗ | 52.2 | +3.6 |
| ✗ | ✗ | ✓ | 44.1 | +11.7 |
| ✓ | ✗ | ✓ | 42.2 | +13.6 |
| LoRA [17] | | | 49.7 | +6.1 |
| DoRA [30] | | | 48.1 | +7.7 |

Table 6: Computational analysis of different methods. "#Param." denotes the number of learnable parameters, while "#Extra Param." refers to additional parameters introduced during CTTA. "#FP" and "#BP" indicate the forward and backward propagation times, respectively. "Time" represents the relative computation time (normalized by TENT).

| Method | #Param. | #Extra Param. | #FP | #BP | Time | Err. Mean↓ |
|---|---|---|---|---|---|---|
| TENT [55] | 0.03M | | 1 | 1 | 1.0 | 51.0 |
| CoTTA [51] | 86.57M | | 11.7 | 1 | 3.6 | 54.8 |
| VDP [9] | 1800 | ✓ | 2 | 1 | 1.5 | 50.0 |
| EcoTTA [51] | 3.46M | ✓ | 1 | 1 | 1.9 | 48.0 |
| ViDA [29] | 7.13M | ✓ | 11 | 1 | 2.8 | 43.4 |
| Ours | 1.24M | ✓ | 1 | 1 | 1.6 | 42.2 |

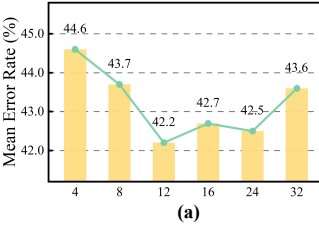 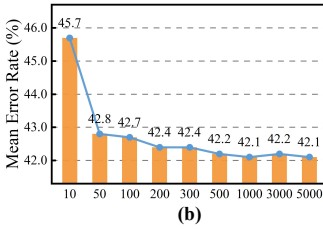 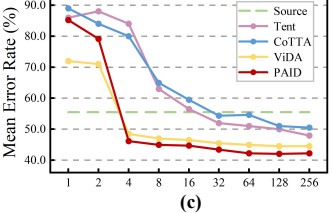

Figure 6: Ablation on (a) the coefficient of the injected orthogonal matrix, (b) the number of source domain samples, and (c) the test-time batch size

Our experiments reveal that the best performance is achieved when the number of matrices is set to 12, indicating an efficient trade-off between capacity and stability.

**Effect of Number of Source Examples.** We investigate the sensitivity of PAID to the amount of source data by varying the number of images used to pre-compute source-domain statistics from 0 to 5,000. As shown in Fig. 6 (b), our method achieves strong performance with as few as 500 source images. It is worth emphasizing that these statistics are computed once prior to CTTA and are not involved in the online adaptation process. Moreover, storing the computed statistics incurs negligible memory cost. These results demonstrate that PAID requires minimal source-domain information to function effectively, making it practical for real-world scenarios.

**Effect of Test Batch Size.** To comprehensively evaluate the impact of test-time batch size, we compare various CTTA methods under batch sizes ranging from 1 to 256. As shown in Fig. 6 (c), a consistent trend emerges across all methods: regardless of their objective functions, performance remains stable with sufficiently large batches but deteriorates as the batch size decreases. In the extreme case of single-sample adaptation, all methods suffer substantial performance drops. In addition, our method maintains decent performance as long as the batch size exceeds 4, and consistently outperforms all comparison methods beyond this threshold.

**Analysis of Computation, Parameter, and Latency.** We analyze the computational complexity of different methods in Tab. 6, comparing the number of learnable parameters, the number of forward and backward passes, and relative runtime. While achieving the best overall performance, our method maintains a relatively small number of learnable parameters and avoids the repeated forward passes required by methods such as CoTTA, VDP, and ViDA. As a result, the increase in computational cost remains moderate, striking a good balance between efficiency and effectiveness.

## 5 Conclusion

This work demonstrates that the pairwise angular structure of source pre-trained weights encodes the domain-invariant semantic prior, supported by statistical analysis, functional validation, visual evidence, intuitive explanation, and theoretical justification. Leveraging this insight, we decompose weights into magnitude and direction, allowing magnitudes to adapt freely while constraining directional updates to global rotations via chained Householder transformations. This preserves the intrinsic angular structure during adaptation. Extensive experiments validate its efficacy.

## Acknowledgements

This work was supported by the National Natural Science Foundation of China (NSFC) under Grants 62225207, 62436008, 62422609 and 62276243.

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

# A Additional Details of Three Motivation Experiments

In Experiment 1, we compute a weighted average of three statistics (mean, variance, and hyperspherical energy) across all linear layers, followed by min-max normalization across different cross-domain settings to produce the visualization in Fig. 1. The TTA results correspond to our proposed method, PAID, applied under a non-continual setting where all linear layers are adapted. The fine-tune baseline refers to supervised fine-tuning of the pret-rained model on the CIFAR-100 dataset for 3 epochs, using warm-up and cosine annealing learning rate schedules. All linear layers and the classification head are learnable during this process. In Experiment 2, the orthogonal rotation is implemented using the chained orthogonal matrices described later in the paper. All hyperparameter settings are aligned with those specified for ImageNet-C in our implementation details. For the attention map visualization in Experiment 3, we use the attention-rollout codebase.[3]

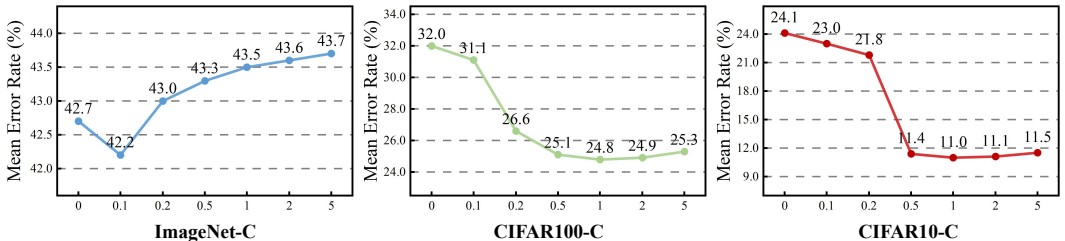

Figure 7: Ablation on loss balancing coefficient $\lambda$

Table 7: Ablation on the selection of linear layer injection

| Ablation | Gaussian | Shot | Impulse | Defocus | Glass | Motion | Zoom | Snow | Frost | Fog | Bright | Contrast | Elastic | Pixel | JPEG | Mean↓ |
|----------|----------|------|---------|---------|-------|--------|------|------|-------|------|--------|----------|---------|-------|------|-------|
| qv | 51.1 | 44.2 | 44.5 | 60.0 | 58.3 | 50.3 | 50.8 | 40.8 | 41.7 | 49.4 | 25.3 | 59.0 | 43.4 | 37.1 | 35.8 | 46.1 |
| qk | 52.3 | 48.5 | 47.2 | 66.2 | 67.7 | 56.9 | 60.1 | 46.1 | 44.7 | 58.8 | 28.1 | 77.8 | 49.8 | 40.1 | 39.4 | 52.2 |
| qkv | 50.7 | 44.5 | 44.2 | 62.8 | 59.1 | 53.0 | 53.5 | 43.4 | 43.4 | 54.1 | 27.0 | 64.1 | 43.8 | 37.5 | 37.3 | 47.9 |
| qkvo | 48.8 | 43.0 | 43.2 | 58.5 | 56.8 | 51.6 | 50.0 | 41.0 | 42.2 | 48.5 | 26.7 | 57.8 | 44.0 | 38.6 | 37.5 | 45.9 |
| qkvom | 48.8 | 43.7 | 44.4 | 49.4 | 49.6 | 47.3 | 44.2 | 37.5 | 39.4 | 42.1 | 25.2 | 50.0 | 39.3 | 35.5 | 36.5 | 42.2 |

Table 8: 10-Round CTTA classification results on ImageNet-C, CIFAR100-C, and CIFAR10-C.

| Dataset | Setting | Gaussian | Shot | Impulse | Defocus | Glass | Motion | Zoom | Snow | Frost | Fog | Bright | Contrast | Elastic | Pixel | JPEG | Mean↓ |
|---------|---------|----------|------|---------|---------|-------|--------|------|------|-------|------|--------|----------|---------|-------|------|-------|
| ImageNet-C | 1-Round | 48.8 | 43.7 | 44.4 | 49.4 | 49.6 | 47.3 | 44.2 | 37.5 | 39.4 | 42.1 | 25.2 | 50.0 | 39.3 | 35.5 | 36.5 | 42.2 |
| | 10-Round | 47.0 | 43.8 | 43.5 | 48.6 | 50.6 | 46.2 | 42.8 | 39.5 | 38.8 | 40.5 | 25.5 | 47.2 | 39.2 | 34.8 | 36.4 | 41.6 |
| CIFAR100-C | 1-Round | 40.7 | 31.9 | 20.4 | 19.8 | 35.9 | 23.0 | 16.3 | 20.5 | 18.2 | 25.3 | 12.6 | 19.8 | 29.4 | 28.2 | 31.3 | 24.9 |
| | 10-Round | 35.5 | 31.9 | 18.6 | 19.5 | 35.7 | 25.1 | 16.7 | 18.1 | 19.3 | 21.5 | 12.1 | 19.5 | 26.2 | 29.7 | 35.2 | 24.3 |
| CIFAR10-C | 1-Round | 22.9 | 11.8 | 9.9 | 9.1 | 16.7 | 10.8 | 7.4 | 7.4 | 6.6 | 11.4 | 4.5 | 9.3 | 12.8 | 9.4 | 14.5 | 11.0 |
| | 10-Round | 15.9 | 13.5 | 9.0 | 9.3 | 15.3 | 9.1 | 8.9 | 7.0 | 6.7 | 10.2 | 4.2 | 8.2 | 15.8 | 9.6 | 12.0 | 10.3 |

# B Additional Ablation Studies

**Effect of Loss Balancing Coefficient.** Unlike the orthogonal matrix coefficient $r$, which shows a consistent optimal value across the three classification benchmarks, the optimal value of the loss balancing coefficient $\lambda$ exhibits outliers. Fig. 7 show that ImageNet-C achieves optimal accuracy with a much smaller $\lambda$ than CIFAR100-C and CIFAR10-C. ImageNet-C keeps its native resolution of $224 \times 224$, and most corruptions simply shift global intensity, so aligning feature means removes most of the domain shift and only a small weight on variance alignment is needed. By contrast, CIFAR images are first enlarged from $32 \times 32$ to $384 \times 384$, which spreads each pixel and magnifies local artifacts, making second-order statistics more important; therefore, a larger $\lambda$ that emphasizes variance alignment is required on the CIFAR variants.

**Effect of Linear Layer Injection.** We examine where to place the orthogonal update in Tab. 7. ViT-Base contains five types of linear layers: q, k, v, o (attention output), and m (MLP). We find that the best performance is achieved when all linear layers are updated (qkvom), indicating that

[3]https://github.com/BoCtrl-C/attention-rollout

Table 9: Classification error rate (%) for ImageNet-to-ImageNet-C, evaluated on ViT-Base with corruption severity level 5. **Bold** indicates the best performance.

| Method | Gaussian | Shot | Impulse | Defocus | Glass | Motion | Zoom | Snow | Frost | Fog | Bright | Contrast | Elastic | Pixel | JPEG | Mean↓ | Gain↑ |
|---|---|---|---|---|---|---|---|---|---|---|---|---|---|---|---|---|---|
| Source | 53.0 | 51.8 | 52.1 | 68.5 | 78.8 | 58.5 | 63.3 | 49.9 | 54.2 | 57.7 | 26.4 | 91.4 | 57.5 | 38.0 | 36.2 | 55.8 | 0.0 |
| Pseudo [24] | **45.2** | **40.4** | **41.6** | 51.3 | 53.9 | 45.6 | 47.7 | 40.4 | 45.7 | 93.8 | 98.5 | 99.9 | 99.9 | 98.9 | 99.6 | 66.8 | -11.0 |
| TENT [55] | 52.2 | 48.9 | 49.2 | 65.8 | 73.0 | 54.5 | 58.4 | 44.0 | 47.7 | 50.3 | 23.9 | 72.8 | 55.7 | 34.4 | 33.9 | 51.0 | +4.8 |
| CoTTA [60] | 52.9 | 51.6 | 51.4 | 68.3 | 78.1 | 57.1 | 62.0 | 48.2 | 52.7 | 55.3 | 25.9 | 90.0 | 56.4 | 36.4 | 35.2 | 54.8 | +1.0 |
| VDP [9] | 52.7 | 51.6 | 50.1 | 58.1 | 70.2 | 56.1 | 58.1 | 42.1 | 46.1 | 45.8 | 23.6 | 70.4 | 54.9 | 34.5 | 36.1 | 50.0 | +5.8 |
| SAR [42] | 45.8 | 45.9 | 47.7 | 52.3 | 63.7 | 46.2 | 50.9 | 40.3 | 42.4 | **41.8** | 24.4 | 53.4 | 53.6 | 38.4 | 36.6 | 45.6 | +10.2 |
| RoTTA [68] | 51.5 | 50.3 | 51.7 | 60.4 | 58.7 | 52.6 | 54.8 | 47.2 | 43.5 | 42.8 | 25.9 | 49.1 | 48.8 | 46.3 | 39.7 | 48.2 | +7.6 |
| EcoTTA [51] | 48.1 | 45.6 | 46.3 | 56.5 | 67.1 | 50.4 | 57.1 | 41.3 | 44.5 | 43.8 | **24.1** | 71.6 | 54.8 | 34.1 | 34.8 | 48.0 | +7.8 |
| ViDA [29] | 47.7 | 42.5 | 42.9 | 52.2 | 56.9 | 45.5 | 48.9 | 38.9 | 42.7 | 40.7 | 24.3 | 52.8 | 49.1 | 33.5 | 32.3 | 43.4 | +12.4 |
| C-MAE [28] | 46.3 | 41.9 | 42.5 | 51.4 | 54.9 | **43.3** | **40.7** | **34.2** | 35.8 | 64.3 | 23.4 | 60.3 | **37.5** | **29.2** | **31.4** | 42.5 | +13.3 |
| Ours | 48.8 | 43.7 | 44.4 | **49.4** | **49.6** | 47.3 | 44.2 | 37.5 | 39.4 | 42.1 | 25.2 | **50.0** | 39.3 | 35.5 | 36.5 | **42.2** | **+13.6** |

Table 10: Classification error rate (%) for CIFAR100-to-CIFAR100-C, evaluated on ViT-Base with corruption severity level 5. **Bold** indicates the best performance.

| Method | Gaussian | Shot | Impulse | Defocus | Glass | Motion | Zoom | Snow | Frost | Fog | Bright | Contrast | Elastic | Pixel | JPEG | Mean↓ | Gain↑ |
|---|---|---|---|---|---|---|---|---|---|---|---|---|---|---|---|---|---|
| Source | 55.0 | 51.5 | 26.9 | 24.0 | 60.5 | 29.0 | 21.4 | 21.1 | 25.0 | 35.2 | 11.8 | 34.8 | 43.2 | 56.0 | 35.9 | 35.4 | 0.0 |
| Pseudo [24] | 53.8 | 48.9 | 25.4 | 23.0 | 58.7 | 27.3 | 19.6 | 20.6 | 23.4 | 31.3 | 11.8 | 28.4 | 39.6 | 52.3 | 33.9 | 33.2 | +2.2 |
| TENT [55] | 53.0 | 47.0 | 24.6 | 22.3 | 58.5 | 26.5 | 19.0 | 21.0 | 23.0 | 30.1 | 11.8 | 25.2 | 39.0 | 47.1 | 33.3 | 32.1 | +3.3 |
| CoTTA [60] | 55.0 | 51.3 | 25.8 | 24.1 | 59.2 | 28.3 | 21.0 | 21.0 | 24.7 | 34.9 | 11.7 | 31.7 | 40.4 | 55.7 | 35.6 | 34.8 | +0.6 |
| VDP [9] | 54.8 | 51.2 | 25.6 | 24.2 | 59.1 | 28.8 | 21.2 | 20.5 | 23.3 | 33.8 | **7.5** | **11.7** | 32.0 | 51.7 | 35.2 | 32.0 | +3.4 |
| ViDA [29] | 50.1 | 40.7 | 22.0 | 21.2 | 45.2 | **21.6** | 16.5 | **17.9** | **16.6** | 25.6 | 11.5 | 29.0 | 29.6 | 34.7 | 27.1 | 27.3 | +8.1 |
| C-MAE [28] | 48.6 | **30.7** | **18.5** | 21.3 | 38.4 | 22.2 | 17.5 | 19.3 | 18.0 | **24.8** | 13.1 | 27.8 | 31.4 | 35.5 | **29.5** | 26.4 | +9.0 |
| Ours | **40.7** | 31.9 | 20.4 | **19.8** | **35.9** | 23.0 | **16.3** | 20.5 | 18.2 | 25.3 | 12.6 | 19.8 | **29.4** | **28.2** | 31.3 | **24.9** | **+10.5** |

distributing the orthogonal correction across both attention and feed-forward paths is essential. This suggests that corruption-induced degradation affects the model in a layer-wise and cumulative manner, and only full-layer adaptation can effectively counteract its impact.

**10-Round Classification CTTA.** To further evaluate the stability and effectiveness of our method, we conduct 10-round CTTA experiments on ImageNet-C, CIFAR100-C, and CIFAR10-C. Specifically, we cycle through the fifteen corruption domains ten times. As shown in Tab. 8, our method remains stable throughout and even achieves slight performance improvements as adaptation progresses. The values reported in the "10-round" row of Tab. 8 represent the average results over the ten rounds.

## C   Fine-grained CTTA Performance

In this section, we extend the classification and segmentation results reported in our submission by presenting a detailed, fine-grained performance analysis. Specifically, we evaluate classification error rate and average mIoU score across distinct corruption types. To complement the summary results shown in Tab. 1 to Tab. 4, we provide additional detailed results in Tab. 9 to Tab. 12. These comprehensive evaluations demonstrate the robustness and effectiveness of our approach in various CTTA scenarios, including ImageNet-to-ImageNet-C, CIFAR-10-to-CIFAR-10-C, CIFAR-100-to-CIFAR-100-C, and Cityscapes-to-ACDC.

## D   Extension to Convolutional Backbones

Beyond Vision Transformers, the core idea behind PAID, i.e., preserving the pairwise angular structure between weight vectors, can be naturally extended to convolutional networks, since each convolutional layer is also a linear operator with a well-defined geometric structure.

In a standard convolutional layer, each output channel corresponds to a filter of shape $(C_{in}, k_h, k_w)$. We flatten this into a vector of length $D = C_{in} \cdot k_h \cdot k_w$, treating each filter as a vector in $\mathbb{R}^D$. Stacking

Table 11: Classification error rate (%) for CIFAR10-to-CIFAR10-C, evaluated on ViT-Base with corruption severity level 5. **Bold** indicates the best performance.

| Method | Gaussian | Shot | Impulse | Defocus | Glass | Motion | Zoom | Snow | Frost | Fog | Bright | Contrast | Elastic | Pixel | JPEG | Mean↓ | Gain↑ |
|---|---|---|---|---|---|---|---|---|---|---|---|---|---|---|---|---|---|
| Source | 60.1 | 53.2 | 38.3 | 19.9 | 35.5 | 22.6 | 18.6 | 12.1 | 12.7 | 22.8 | 5.3 | 49.7 | 23.6 | 24.7 | 23.1 | 28.1 | 0.0 |
| Pseudo [24] | 59.8 | 52.5 | 37.2 | 19.8 | 35.2 | 21.8 | 17.6 | 11.6 | 12.3 | 20.7 | 5.0 | 41.7 | 21.5 | 25.2 | 22.1 | 26.9 | +1.2 |
| TENT [55] | 57.7 | 56.3 | 29.4 | 16.2 | 35.3 | 16.2 | 12.4 | 11.0 | 11.6 | 14.9 | 4.7 | 22.5 | 15.9 | 29.1 | 19.5 | 23.5 | +4.6 |
| CoTTA [60] | 58.7 | 51.3 | 33.0 | 20.1 | 34.8 | 20.0 | 15.2 | 11.1 | 11.3 | 18.5 | 4.0 | 34.7 | 18.8 | 19.0 | 17.9 | 24.6 | +3.5 |
| VDP [9] | 57.5 | 49.5 | 31.7 | 21.3 | 35.1 | 19.6 | 15.1 | 10.8 | 10.3 | 18.1 | 4.0 | 27.5 | 18.4 | 22.5 | 19.9 | 24.1 | +4.0 |
| ViDA [29] | 52.9 | 47.9 | 19.4 | 11.4 | 31.3 | 13.3 | 7.6 | 7.6 | 9.9 | 12.5 | **3.8** | 26.3 | 14.4 | 33.9 | 18.2 | 20.7 | +7.4 |
| C-MAE [28] | 30.6 | 18.9 | 11.5 | 10.4 | 22.5 | 13.9 | 9.8 | 6.6 | **6.5** | **8.8** | 4.0 | **8.5** | **12.7** | **9.2** | 14.4 | 12.6 | +15.5 |
| Ours | **22.9** | **11.8** | **9.9** | **9.1** | **16.7** | **10.8** | **7.4** | 7.4 | 6.6 | 11.4 | 4.5 | 9.3 | 12.8 | 9.4 | 14.5 | **11.0** | **+17.1** |

Table 12: Average mIoU score (%) for Cityscapes-to-ACDC, evaluated on Segformer-B5. The same target domains are repeated three rounds. **Bold** indicates the best performance.

| Method | Round 1 | | | | | Round 2 | | | | | Round 3 | | | | | Mean↑ | Gain↑ |
|---|---|---|---|---|---|---|---|---|---|---|---|---|---|---|---|---|---|
| | Fog | Night | Rain | Snow | Mean↑ | Fog | Night | Rain | Snow | Mean↑ | Fog | Night | Rain | Snow | Mean↑ | | |
| Source | 69.1 | 40.3 | 59.7 | 57.8 | 56.7 | 69.1 | 40.3 | 59.7 | 57.8 | 56.7 | 69.1 | 40.3 | 59.7 | 57.8 | 56.7 | 56.7 | 0.0 |
| TENT [55] | 69.0 | 40.2 | 60.1 | 57.3 | 56.7 | 68.3 | 39.0 | 60.1 | 56.3 | 55.9 | 67.5 | 37.8 | 59.6 | 55.0 | 55.0 | 55.7 | -1.0 |
| CoTTA [60] | 70.9 | 41.2 | 62.4 | 59.7 | 58.6 | 70.9 | 41.1 | 62.6 | 59.7 | 58.6 | 70.9 | 41.0 | 62.7 | 59.7 | 58.6 | 58.6 | +1.9 |
| DePT [10] | 71.0 | 40.8 | 58.2 | 56.8 | 56.5 | 68.2 | 40.0 | 55.4 | 53.7 | 54.3 | 66.4 | 38.0 | 47.3 | 47.2 | 49.7 | 53.4 | -3.3 |
| VDP [9] | 70.5 | 41.1 | 62.1 | 59.5 | 58.3 | 70.4 | 41.1 | 62.2 | 59.4 | 58.2 | **70.4** | 41.0 | 62.2 | 59.4 | 58.2 | 58.2 | +1.5 |
| SAR [42] | 69.0 | 40.2 | 60.1 | 57.3 | 56.7 | 69.0 | 40.3 | 60.0 | **67.8** | 59.3 | 67.5 | 37.8 | 59.6 | 55.0 | 55.0 | 57.0 | +0.3 |
| EcoTTA [51] | 68.5 | 35.8 | 62.1 | 57.4 | 56.0 | 68.3 | 35.5 | 62.3 | 57.4 | 55.9 | 68.1 | 35.3 | 62.3 | 57.3 | 55.8 | 55.8 | -0.9 |
| ViDA [29] | 71.6 | 43.2 | 66.0 | **63.4** | 61.1 | **73.2** | 44.5 | 67.0 | 63.9 | 62.2 | 73.2 | 44.6 | 67.2 | **64.2** | 62.3 | 61.9 | +5.2 |
| C-MAE [28] | **71.9** | 44.6 | 67.4 | 63.2 | **61.8** | 71.7 | 44.9 | 66.5 | 63.1 | 61.6 | 72.3 | 45.4 | 67.1 | 63.1 | 62.0 | 61.8 | +5.1 |
| Ours | 69.6 | **45.5** | **68.0** | 60.7 | 61.0 | 72.3 | **45.2** | 66.8 | 62.5 | **61.7** | 72.6 | **46.9** | **68.4** | 63.8 | **62.9** | 62.2 | **+5.5** |

these column-wise yields a weight matrix $W \in \mathbb{R}^{D \times C_\text{out}}$, which we decompose as:

$$W = M \odot \hat{W},$$

where $\hat{W}$ contains unit-norm columns and $M \in \mathbb{R}^{1 \times C_\text{out}}$ stores the magnitudes. During adaptation, we freeze $\hat{W}$ and update $M$ along with a learnable orthogonal matrix $O \in \mathbb{R}^{C_\text{out} \times C_\text{out}}$, constructed via Householder reflections. The updated weights become:

$$W' = M \odot \left( \hat{W} \cdot O \right).$$

Here, $O$ operates along the output channel dimension. The spatial structure within each filter remains intact because we do not rotate across spatial positions. Notably, flattening convolution filters and treating them as vectors is a standard practice in low-rank adaptation and model compression literature. Reshaping back after rotation ensures that spatial locality is fully preserved during convolution.

To validate this extension, we follow standard CTTA setups used in prior works: WideResNet-28 on CIFAR-10-C, ResNeXt-29 on CIFAR-100-C, and ResNet-50 on ImageNet-C. We select representative and open-sourced baselines for comparison, using benchmark results that have been consistently reported and validated across multiple CTTA studies. As shown in Tab. 13, PAID achieves consistent performance gains across all convolutional backbones, indicating that PAID generalizes beyond ViTs and is also applicable to CNNs.

To further explain why PAID remains valid in convolutional networks, we revisit several prior studies on the geometric structure of convolutional filters. DCNet [33] shows that the convolution operation can be reformulated as an inner product and decomposed into magnitude and angular: the magnitude better models intra-class variation, while the angular captures semantic differences. SphereConv [34] further demonstrates that learning only the angular component of convolutional filters is sufficient for semantic classification. These findings indicate that the angular structure in CNNs also carries domain-invariant semantics. Therefore, applying PAID's strategy to convolutional networks is not only formally valid, but also empirically supported by prior work.

Table 13: Extension of PAID to convolutional backbones. Test error (%) on three CTTA benchmarks.

| Dataset | Model | Source | TENT [55] | CoTTA [60] | AdaCon [1] | CRG [79] | LAW [44] | **Ours** |
|---|---|---|---|---|---|---|---|---|
| ImageNet-C | ResNet-50 | 82.0 | 62.6 | 62.7 | 65.5 | 59.1 | 60.1 | **58.4** |
| CIFAR100-C | ResNeXt-29 | 46.4 | 60.9 | 32.5 | 33.4 | 29.0 | 30.9 | **28.1** |
| CIFAR10-C | WideResNet-28 | 43.5 | 20.7 | 16.1 | 18.5 | 15.9 | 15.7 | **15.0** |

# E  Intuition Behind the Performance Drop with Small Batch Sizes

As shown in Fig. 6 (c), most existing CTTA methods, including ours, exhibit a notable performance drop as the batch size decreases. This behavior is not specific to our approach, similar trends have been reported in prior works such as [68] (Fig. 4(d)) and [51] (Tab. 5), where small-batch scenarios consistently lead to degraded adaptation performance.

For our method in particular, the primary factor is the reliability of target feature statistics. In Eq. 9, we perform alignment between source and target distributions using the mean and standard deviation computed from the current test batch. When the batch size is small, these estimates become highly variable and noisy, which results in unstable gradients and impairs the optimization process. This instability ultimately hinders adaptation effectiveness.

In contrast, larger batch sizes produce more reliable and smoother estimates of distributional statistics, effectively reducing variance in the adaptation signal. This leads to more stable parameter updates and improved learning dynamics. Empirically, we observe that performance becomes acceptable when the batch size exceeds 4, and saturates when the batch size is greater than 32.

# F  Limitations

While PAID shows consistent gains under diverse corruptions, it is built upon a central hypothesis: the angular structure learned by source pre-trained models is potentially generalizable across diverse target domains. This assumption has been preliminarily validated through experiments, yet its applicability boundaries remain unclear, particularly under extreme conditions. Moreover, hyper-parameter tuning remains a common challenge in the CTTA community, underscoring the need for automated tuning strategies to enhance practical usability. Finally, extending the idea of angular structure preservation to tasks such as few-shot learning, and continual learning presents a promising direction, where maintaining the geometric stability of weights may likewise prove essential.

