# OpenReview forum: "PAID: Pairwise Angular-Invariant Decomposition for Continual Test-Time Adaptation"
_NeurIPS.cc/2025/Conference — NeurIPS 2025 poster_

### Official Review · Reviewer_tGtE · 2025-06-29

**Clarity:** 3
**Significance:** 3
**Originality:** 2
**Rating:** 4
**Confidence:** 3

**Summary:**

The authors focus on the often-overlooked pre-trained weights and empirically discover that the pairwise angular structure among weight vectors remains stable across a wide range of corrupted domains, capturing domain-invariant semantic information. Motivated by this finding, they propose PAID (Pairwise Angular-Invariant Decomposition), a method that adapts the model during test time by updating only the weight magnitudes and applying a learnable orthogonal matrix—constructed via Householder transformations—to preserve the relative angular structure. Extensive experiments on four widely-used CTTA benchmarks validate the effectiveness and generalizability of the proposed approach.

**Questions:**

1. Can the proposed method be applied to other network architectures beyond Transformers?

**Ethical Concerns:**

["NO or VERY MINOR ethics concerns only"]

**Final Justification:**

The authors’ additional experiments adequately resolve my concerns and I have no other questions. So I retain my original score.

**Limitations:**

yes

**Quality:**

3

**Strengths And Weaknesses:**

**Strengths:**

1. The paper tackles the TTA problem from a novel and insightful perspective, focusing on the geometric structure of pre-trained weights.
2. The writing is generally clear, and the authors effectively convey their motivation through well-designed experiments and illustrative figures.
3. The experimental evaluation is relatively comprehensive, covering multiple classification and segmentation benchmarks with thorough ablation studies.

**Weaknesses:**

1. The method is only evaluated on Transformer-based architectures. Its applicability to other network types, such as CNNs, remains unexplored and should be discussed or validated.
2. Some of the text in Figure 5 is blurry.

---

> ### Author Rebuttal · Authors · 2025-07-31
>
> Thank you for your valuable and constructive comments. We appreciate your recognition of our novel and insightful perspective on CTTA based on the geometric structure of pre-trained weights, as well as your positive feedback on the clarity of our writing, the motivation conveyed through well-designed experiments and figures, and the comprehensive evaluation across multiple benchmarks with detailed ablation studies. Below, we address your concerns and questions individually:
>
> ## **Q1: Extension to other network architectures beyond Transformer**
> Thank you for your forward-looking suggestion. The key insight of PAID, preserving the pairwise angular structure between weight vectors to retain domain-invariant semantics, can be **formally** extended to convolutional networks, where each convolutional layer is also a parameterized linear transformation.
>
> In a standard convolutional layer, each output channel corresponds to a filter of shape $(C_{\text{in}}, k_h, k_w)$. We flatten this into a vector of length $D = C_{\text{in}} \cdot k_h \cdot k_w$, treating each filter as a vector in $\mathbb{R}^D$. Stacking these column-wise yields a weight matrix $W \in \mathbb{R}^{D \times C_{\text{out}}}$, which we decompose as:
>
> $
> W = M \odot \hat{W},
> $
>
> where $\hat{W}$ contains unit-norm columns and $M \in \mathbb{R}^{1 \times C_{\text{out}}}$ stores the magnitudes. During adaptation, we freeze $\hat{W}$ and update $M$ along with a learnable orthogonal matrix $O \in \mathbb{R}^{C_{\text{out}} \times C_{\text{out}}}$, constructed via Householder reflections. The updated weights become:
>
> $
> W' = M \odot (\hat{W} \cdot O).
> $
>
> Here, $O$ operates along the output channel dimension. The spatial structure within each filter remains intact because we do not rotate across spatial positions.
> Notably, flattening convolution filters and treating them as vectors is a standard practice in low-rank adaptation and model compression literature. Reshaping back after rotation ensures that spatial locality is fully preserved during convolution.
>
> To **validate** this extension, we follow standard CTTA setups used in prior works: WideResNet-28 on CIFAR-10-C, ResNeXt-29 on CIFAR-100-C, and ResNet-50 on ImageNet-C. We select representative and open-sourced baselines for comparison, using benchmark results that have been consistently reported and validated across multiple CTTA studies. The results show **consistent performance gains** across all convolutional backbones, indicating that PAID generalizes beyond ViTs and is also applicable to CNNs.
>
> | Dataset     | Model          | Source | TENT[31]  | CoTTA[9] | AdaCon[36] | CRG(1) | LAW(2) | Ours |
> |-------------|----------------|--------|-------|-------|--------|--------|------|------|
> | ImageNet-C  | ResNet-50       | 82.0   | 62.6  | 62.7  | 65.5   | 59.1   | 60.1   | **58.4** |
> | CIFAR100-C   | ResNeXt-29    | 46.4   | 60.9  | 32.5  | 33.4   | 29.0   | 30.9   | **28.1** |
> | CIFAR10-C  | WideResNet-28   | 43.5   | 20.7  | 16.1  | 18.5   | 15.9   | 15.7   | **15.0** |
>
> To further **explain** why PAID remains valid in convolutional networks, we revisit several prior studies on the geometric structure of convolutional filters. DCNet [18] shows that the convolution operation can be reformulated as an inner product and decomposed into magnitude and angular: the magnitude better models intra-class variation, while the angular captures semantic differences. SphereConv [15] further demonstrates that learning only the angular component of convolutional filters is sufficient for semantic classification. These findings indicate that the angular structure in CNNs also carries domain-invariant semantics. Therefore, applying PAID’s strategy to convolutional networks is not only formally valid, but also **empirically supported** by prior work.
>
> ## **Q2: Blurry text in Figure 5**
> Thank you very much for pointing out this typo. We will replace the figure with a clearer one and use a larger font in the revision.
>
> (1) Reshaping the Online Data Buffering and Organizing Mechanism for Continual Test-Time
> Adaptation, ECCV 2024.
>
> (2) Layer-wise Auto-Weighting for Non-Stationary Test-Time Adaptation, WACV 2024.

---

> > ### Comment · Reviewer_tGtE · 2025-08-04
> >
> > Thank you for the clarification. I will keep my original score.

---

> > > ### Author Response · Authors · 2025-08-05
> > >
> > > Thank you for your constructive follow-up and for taking the time to carefully review our rebuttal. You have raised two important points: extending the evaluation to a wider range of backbone architectures and improving the clarity of the figures in the manuscript. We are grateful for these suggestions, as they both address aspects that can enhance the accessibility and generality of our work.
> > >
> > > We recognize that evaluating our methodacross diverse backbone architectures is essential for demonstrating the applicability of our method. In the revised version, we will include additional experiments on different backbones to provide stronger evidence of the method’s generality. Regarding the figures, we fully agree that visual clarity is crucial for effectively communicating our ideas and results. In the updated manuscript, we will revise the illustrations to ensure that all figures are clearer, more informative, and easier to interpret, so that readers can better follow the technical content.
> > >
> > > It is our sincere hope that this work can introduce a new design paradigm and inspiration for the CTTA community. In contrast to most existing methods that focus on exploiting target stream data, our work is the first to treat pre-trained weights as more than static initializations. This enables a structured update process that preserves the semantic organization encoded during pre-training. We also believe that the insights presented here have the potential to be extended to other areas, such as transfer learning and knowledge retention, thereby contributing more broadly to the community.
> > >
> > > We greatly appreciate your thoughtful comments. Thank you once again for your valuable engagement.

---

### Official Review · Reviewer_e8tG · 2025-06-30

**Clarity:** 4
**Significance:** 3
**Originality:** 3
**Rating:** 4
**Confidence:** 4

**Summary:**

This paper presents a new approach for Continual Test-Time Adaptation (CTTA) by analyzing and leveraging the geometric properties of pre-trained neural network weights. The authors find that the pairwise angular structure of weights is stable across different domains and encodes domain-invariant semantic information, whereas magnitude and absolute angle are more domain-specific. Based on this insight, they propose PAID, a method that adapts only the magnitudes and applies a global orthogonal transformation to directions, strictly preserving the pairwise angular structure. The method consistently outperforms state-of-the-art baselines on several classification and segmentation benchmarks.

**Questions:**

1. Can the authors provide additional theoretical justification for why pairwise angular structure is robust to domain-specific shifts but sensitive to semantic changes?

2. Have the authors investigated whether the pairwise angular invariance observed in ViT models also holds for convolutional layers or non-transformer architectures? If not, could some pilot experiments or discussion be provided to address the generality of the method?

**Ethical Concerns:**

["NO or VERY MINOR ethics concerns only"]

**Final Justification:**

The authors have addressed my concerns, including clarifying the scope of their approach beyond ViT-based models and providing additional theoretical foundations. Given these clarifications, and the overall strength of the paper's motivation and experimental design, I will maintain my original rating.

**Limitations:**

Yes, the authors have adequately discussed the limitations of proposed method in the supplementary material.

**Quality:**

4

**Strengths And Weaknesses:**

Strengths:

1. The key empirical finding that the pairwise angular structure of pre-trained weights is a domain-invariant semantic prior provides a fresh direction for CTTA.

2. The approach is well-motivated. The motivation is supported by comprehensive experiments (statistical analysis, functional ablation, visualization), leading directly to the method’s design.

3. The paper is well-organized and clearly written, with intuitive explanations and detailed appendices.

Weaknesses:

1. Limited Theoretical Analysis. While the empirical evidence for pairwise angular structure invariance is strong, the theoretical basis for why it encodes domain-invariant semantics is largely motivated by analogy and experiment rather than formal analysis.

2. The method is evaluated primarily on ViT architectures and transformer-based segmentation models. It is unclear whether the same invariance properties and adaptation mechanisms apply as strongly to convolutional or non-transformer architectures.

3. Typo in Figure 5: wectors.

---

> ### Author Rebuttal · Authors · 2025-07-31
>
> Thank you for your valuable and constructive comments. We appreciate your recognition of our key empirical finding, as well as your positive feedback on our well-motivated approach, the comprehensive supporting experiments, and the clarity and organization of the paper. Below, we address your concerns and questions individually:
>
> ## **Q1: Theoretical analysis for why pairwise angular structure is robust to domain-specific shifts but sensitive to semantic changes**
> To address your question, we provide a formal analysis using a simplified linear classification model. Specifically, we prove that:
>
> - **As the domain-specific shift applies the same transformation to all input samples regardless of class**, the optimal classifier in the target domain can be written as a shared linear transformation of the source classifier, under mild and mathematically defined conditions. This transformation preserves all pairwise angles between class weights.
> - **In contrast, semantic changes induce class-specific input transformations**, which break this shared structure and necessarily alter the pairwise angular geometry.
>
> This supports the core design of PAID: using a orthogonal rotation and magnitude adjustment is sufficient for adapting to domain-specific corruptions, but insufficient for adapting to semantic shift, where the angular structure must be re-learned. We now present the mathematical proof in detail.
>
> ---
>
> ### Mathematical Formulation
>
> #### Linear Classification Model
>
> Let $ x \in \mathbb{R}^a $ be the input vector, and for each class $ c \in \{1, ..., C\} $, let $ w_c \in \mathbb{R}^a $ be the classifier weight vector.  The predicted class is:
>
> $
> \hat{y}(x) = \arg\max_{c} w_c^\top x.
> $
>
> We compare two settings:
>
> ---
>
> ### Case 1: Domain-specific Degradation (Class-agnostic input transformation)
>
> We assume the degraded input is obtained by applying the same transformation to all inputs:
>
> $
> x' = d(x),
> $
>
> and that $ d $ is differentiable near each $ x $. We approximate $ d(x) $ by its first-order Taylor expansion around clean data:
>
> $
> d(x) \approx Sx + q,
> $
>
> where $ S \in \mathbb{R}^{a \times a} $ is the Jacobian matrix at $ x $, assumed to be invertible and independent of class $ y $.  We formalize this assumption as:
>
> > **Assumption 1 (Class-agnostic linear perturbation):**
> > There exists an invertible matrix $ S \in \mathrm{GL}(a) $ such that for all $ x $,  $\quad d(x) = Sx + q, \quad \text{with } S \text{ independent of class } y$.
>
> We define the optimal classifier in the target domain as the one that operates on $ x' = Sx $ but maintains the same class boundaries in the original space. This gives:
>
> $
> w'_c = S^{-\top} w_c, \quad \forall c.
> $
>
> Now we examine the effect of this transformation on pairwise angles. We perform the polar decomposition:
>
> $
> S^{-\top} = RH,
> $
>
> where $ R \in \mathbb{R}^{a \times a} $ is orthogonal $( R^\top R = I )$, and $ H \in \mathbb{R}^{a \times a} $ is symmetric positive definite. We now examine the transformed weights:
>
> $
> \tilde{w}_c = RH w_c.
> $
>
> We compute the cosine similarity between any two transformed weights:
>
> $
> \cos \angle(\tilde{w}_c, \tilde{w}_d)
> = \frac{(RH w_c)^\top (RH w_d)}{\|RH w_c\| \cdot \|RH w_d\|}
> = \frac{w_c^\top H^2 w_d}{\|H w_c\| \cdot \|H w_d\|}.
> $
>
> To ensure angles are preserved, we introduce:
>
> > **Assumption 2 (Near-isotropic perturbation):**
> > The matrix $ H $ satisfies $ H^2 \approx \gamma^2 I $ for some scalar $ \gamma > 0 $.
> > This holds when the degradation has approximately equal effect in all directions (e.g., blur, light noise, JPEG), and ensures angle preservation:
>
> $
> \cos \angle(\tilde{w}_c, \tilde{w}_d) \approx \cos \angle(w_c, w_d).
> $
>
> Thus, under Assumptions 1 and 2, **pairwise angular structure remains invariant under domain-specific shifts**.
>
> ---
>
> ### Case 2: Semantic Change (Class-dependent input transformation)
>
> Suppose the semantic shift modifies each class's input differently:
>
> $
> x' = d_y(x), \quad \text{with } d_y(x) \approx S_y x + q_y,
> $
>
> where $ S_y \in \mathrm{GL}(a) $ varies with class $ y $. Then the optimal classifier becomes:
>
> $
> w''_c = S _{y=c} ^{-\top} w_c.
> $
>
> In this case, each weight undergoes a different transformation. Unless all $ S_y $ are identical up to scaling or rotation, **there exists no single matrix** $ T \in \mathbb{R}^{a \times a} $ such that $ w''_c = T w_c $ for all $ c $. Hence:
>
> There is no shared transformation that preserves the entire angular structure. The pairwise angle matrix must differ from that in the original domain. This proves that **semantic changes necessarily break the pairwise angular structure**.
>
> ---
>
> ### Final Conclusion
>
> - When domain shifts are class-agnostic and locally differentiable, they can be represented by a shared invertible matrix $ S $, and the optimal classifier corresponds to applying the inverse transpose $ S^{-\top} $ to all weights.
> - This transformation preserves angles up to a near-isotropic assumption, justifying why **PAID only needs to learn a shared orthogonal rotation and magnitude adjustment**.
> - When the shift is class-dependent, this shared structure no longer exists, and angular geometry must change.
>
> This provides a direct theoretical justification for your question and explains the observations of our paper.
>
> ## **Q2: Extension to convolutional or non-transformer architectures**
> Thank you for your forward-looking suggestion. The key insight of PAID, preserving the pairwise angular structure between weight vectors to retain domain-invariant semantics, can be **formally** extended to convolutional networks, where each convolutional layer is also a parameterized linear transformation.
>
> In a standard convolutional layer, each output channel corresponds to a filter of shape $(C_{\text{in}}, k_h, k_w)$. We flatten this into a vector of length $D = C_{\text{in}} \cdot k_h \cdot k_w$, treating each filter as a vector in $\mathbb{R}^D$. Stacking these column-wise yields a weight matrix $W \in \mathbb{R}^{D \times C_{\text{out}}}$, which we decompose as:
>
> $
> W = M \odot \hat{W},
> $
>
> where $\hat{W}$ contains unit-norm columns and $M \in \mathbb{R}^{1 \times C_{\text{out}}}$ stores the magnitudes. During adaptation, we freeze $\hat{W}$ and update $M$ along with a learnable orthogonal matrix $O \in \mathbb{R}^{C_{\text{out}} \times C_{\text{out}}}$, constructed via Householder reflections. The updated weights become:
>
> $
> W' = M \odot (\hat{W} \cdot O).
> $
>
> Here, $O$ operates along the output channel dimension. The spatial structure within each filter remains intact because we do not rotate across spatial positions.
> Notably, flattening convolution filters and treating them as vectors is a standard practice in low-rank adaptation and model compression literature. Reshaping back after rotation ensures that spatial locality is fully preserved during convolution.
>
> To **validate** this extension, we follow standard CTTA setups used in prior works: WideResNet-28 on CIFAR-10-C, ResNeXt-29 on CIFAR-100-C, and ResNet-50 on ImageNet-C. We select representative and open-sourced baselines for comparison, using benchmark results that have been consistently reported and validated across multiple CTTA studies. The results show **consistent performance gains** across all convolutional backbones, indicating that PAID generalizes beyond ViTs and is also applicable to CNNs.
>
> | Dataset     | Model          | Source | TENT[31]  | CoTTA[9] | AdaCon[36] | CRG(1) | LAW(2) | Ours |
> |-------------|----------------|--------|-------|-------|--------|--------|------|------|
> | ImageNet-C  | ResNet-50       | 82.0   | 62.6  | 62.7  | 65.5   | 59.1   | 60.1   | **58.4** |
> | CIFAR100-C   | ResNeXt-29    | 46.4   | 60.9  | 32.5  | 33.4   | 29.0   | 30.9   | **28.1** |
> | CIFAR10-C  | WideResNet-28   | 43.5   | 20.7  | 16.1  | 18.5   | 15.9   | 15.7   | **15.0** |
>
> To further **explain** why PAID remains valid in convolutional networks, we revisit several prior studies on the geometric structure of convolutional filters. DCNet [18] shows that the convolution operation can be reformulated as an inner product and decomposed into magnitude and angular: the magnitude better models intra-class variation, while the angular captures semantic differences. SphereConv [15] further demonstrates that learning only the angular component of convolutional filters is sufficient for semantic classification. These findings indicate that the angular structure in CNNs also carries domain-invariant semantics. Therefore, applying PAID’s strategy to convolutional networks is not only formally valid, but also **empirically supported** by prior work.
>
> ## **Q3: Typo in Figure 5**
> Thank you very much for pointing out this typo. We will correct this error in the revision.
>
> (1) Reshaping the Online Data Buffering and Organizing Mechanism for Continual Test-Time
> Adaptation, ECCV 2024.
>
> (2) Layer-wise Auto-Weighting for Non-Stationary Test-Time Adaptation, WACV 2024.

---

> > ### Comment · Reviewer_e8tG · 2025-08-04
> >
> > Thanks for your response. It has solved my concerns. I will maintain my original score.

---

> > > ### Author Response · Authors · 2025-08-05
> > >
> > > Thank you very much for your encouraging follow-up and for engaging with our rebuttal. You have highlighted three key points: providing a thorough theoretical analysis, extending the method to additional backbones, and correcting typographical errors. We truly appreciate these suggestions, particularly the one on strengthening the theoretical foundation. In the revised version, we will incorporate the theoretical analysis to complement the extensive empirical studies, thereby reinforcing the insights of our work from both perspectives.
> > >
> > > It is our sincere hope that this work can introduce a new design paradigm and inspiration for the CTTA community. In contrast to most existing methods that focus on exploiting target stream data, our work is the first to treat pre-trained weights as more than static initializations. This enables a structured update process that preserves the semantic organization encoded during pre-training. We also believe that the insights presented here have the potential to be extended to other areas, such as transfer learning and knowledge retention, thereby contributing more broadly to the community.
> > >
> > > Your constructive feedback has been invaluable in improving our work. Thank you again for your thoughtful engagement.

---

### Official Review · Reviewer_jqAQ · 2025-07-02

**Clarity:** 3
**Significance:** 3
**Originality:** 3
**Rating:** 4
**Confidence:** 4

**Summary:**

This paper uses the geometric properties of pre-trained weights for the task of Continual Test-Time Adaptation (CTTA). The key insight is that the pairwise angular structure of weight vectors is domain-invariant and encodes important semantic information. Based on this, the paper proposes Pairwise Angular-Invariant Decomposition (PAID) that decomposes the weights of linear layers into magnitude and direction. During adaptation, it keeps the original direction vectors frozen but allows their magnitudes to be updated. PAID also introduces a learnable orthogonal matrix, parameterized by Householder reflections, which applies a global rotation to all direction vectors simultaneously. The method is evaluated on several benchmarks for image classification and segmentation and shows consistent performance improvements over existing methods.

**Questions:**

1. The performance drop with small batch sizes is significant. Are there any intuitions?
2. How does PAID perform on CNN architectures like ResNet?
3. The proposed method is sensitive to hyperparameter. Did you also tune parameters for baselines to achieve the best performance for each of them?

**Ethical Concerns:**

["NO or VERY MINOR ethics concerns only"]

**Final Justification:**

The author addressed all my concerns and I am satisfied with the answer. The paper can bring interesting insights to CTTA domain.

**Limitations:**

yes

**Paper Formatting Concerns:**

no major formatting issues

**Quality:**

3

**Strengths And Weaknesses:**

Strengths
1. The paper addresses the Continual Test-Time Adaptation (CTTA) problem, which is a challenging and practical scenario.
2. The paper is well written and easy to follow.
3. The paper provides extensive experiments (ImageNet-C, CIFAR-10/100-C, ACDC), showing the effectiveness and versatility of the proposed method.
4. Although the idea of decomposing weights into magnitude and direction has been explored in other contexts (e.g., DoRA, OFT), it is relatively new in CTTA domain.


Weaknesses
1. The proposed method needs a batch of data for adaptation. As shown in Figure 6 (c), performance degrades significantly with smaller batch sizes, especially for single-sample adaptation.
2. The learning rate and lambda in eq (9) vary across datasets, suggesting that manual tuning is needed for new datasets and tasks, which could be a practical limitation.
3. The method is limited to ViT backbone and not validated on CNNs.
4. The improvement is somehow limited compare to the best baseline. For instance, only 0.3 gain in Table 1 compared with C-MAE and only 0.3 gain in Table 4 compared with ViDA.

---

> ### Author Rebuttal · Authors · 2025-07-31
>
> Thank you for your valuable and constructive comments. We appreciate your recognition of the importance of the CTTA setting, the clarity of our writing, the thorough experimental validation across multiple benchmarks, and the novelty of applying weight decomposition in the CTTA context. Below, we address your concerns and questions individually:
>
> ## **Q1: What is the intuition behind the performance drop with small batch sizes**
>
> As shown in Figure 6 (c), most existing CTTA methods, including ours, exhibit a notable performance drop as the batch size decreases. This behavior is not specific to our approach, similar trends have been reported in prior works such as [24] (Figure 4 (d)) and [28] (Table 5), where small-batch scenarios consistently lead to degraded adaptation performance.
>
> For our method in particular, the primary factor is **the reliability of target feature statistics**. In Equation (9), we perform alignment between source and target distributions using the mean and standard deviation computed from the current test batch. When the batch size is small, these estimates become highly variable and noisy, which results in unstable gradients and impairs the optimization process. This instability ultimately hinders adaptation effectiveness.
>
> In contrast, larger batch sizes produce more reliable and smoother estimates of distributional statistics, effectively reducing variance in the adaptation signal. This leads to more stable parameter updates and improved learning dynamics. **Empirically, we observe that performance becomes acceptable when the batch size exceeds 4, and saturates when the batch size is greater than 32.**
>
> To mitigate this limitation, we also explore a simple enhancement: applying an Exponential Moving Average (EMA) to the target statistics. This temporal smoothing helps stabilize the alignment objective under small-batch regimes and partially alleviates the performance drop when the batch size is 1 or 2. The corresponding results are provided below:
>
> | Batch Size | w/o EMA | w/ EMA |
> |------------|---------|--------|
> | 1          |   85.0    |  69.2    |
> | 2          |   78.7    |  65.1    |
>
> ## **Q2: Clarification and Discussion on the Fairness of Baseline Comparisons and Hyperparameter Tuning**
>
> We sincerely thank the reviewer for raising this practically relevant concern, which prompted us to further reflect on the challenges faced by the CTTA community in real-world deployment.
>
> ### Clarification on the fairness of baseline comparisons
> The baseline results reported in our experiments are all based on benchmarks widely adopted in prior CTTA studies. These results have not only been repeatedly validated in multiple papers, but also follow the hyperparameter settings **specified** in the original papers or open-source code provided by the respective authors. For example, ViDA uses a learning rate of 1e-4 on CIFAR-C, but 5e-7 on ImageNet-C; C-MAE adopts 1e-5 on CIFAR-C, while using 1e-3 on ImageNet-C. Therefore, we ensure that all baseline comparisons are conducted under reasonable and recommended settings, which guarantees the fairness and reliability of our experimental results.
>
> ### Discussion on hyperparameter sensitivity and manual tuning
> We agree with the reviewer that manual tuning may present a limitation in real-world deployment. In fact, sensitivity to hyperparameters such as learning rate and regularization coefficients is a widely recognized challenge across CTTA community, as discussed in the prior work (1). We believe that improving the robustness of CTTA methods to hyperparameters, or exploring automated tuning strategies, is one of the most important directions for future research.
>
> That said, we also note that manual tuning still holds practical value at the current stage and offers useful empirical guidance for deployment. Below we share our observation regarding hyperparameter selection in our method:
>
> - **Loss balancing coefficient λ**: We found that image resolution significantly affects statistical variations. For low-resolution images (e.g., CIFAR with original size 32×32), upsampling to 384×384 amplifies local noise, making variance alignment more important. For high-resolution images (e.g., ImageNet-C with native resolution 224×224), global intensity shifts dominate, so aligning feature means becomes more crucial. This observation provides an empirical basis for choosing λ under different pre-processing conditions.
>
> In addition, we observed that when using the same learning rate settings for both CIFAR-C and ImageNet-C, our method still achieves competitive performance. This demonstrates that our method does not rely heavily on excessive tuning.
>
> | lr      | CIFAR10-C | CIFAR100-C | ImageNet-C |
> |---------|-----------|------------|------------|
> | 8e-5    |   *11.0*    |      *24.9*   |     42.5    |
> | 2e-4    |   11.5    |      24.9   |     *42.2*    |
>
>
>
> ## **Q3: Extension to convolutional networks**
> Thank you for your forward-looking suggestion. The key insight of PAID, preserving the pairwise angular structure between weight vectors to retain domain-invariant semantics, can be **formally** extended to convolutional networks, where each convolutional layer is also a parameterized linear transformation.
>
> In a standard convolutional layer, each output channel corresponds to a filter of shape $(C_{\text{in}}, k_h, k_w)$. We flatten this into a vector of length $D = C_{\text{in}} \cdot k_h \cdot k_w$, treating each filter as a vector in $\mathbb{R}^D$. Stacking these column-wise yields a weight matrix $W \in \mathbb{R}^{D \times C_{\text{out}}}$, which we decompose as:
>
> $
> W = M \odot \hat{W},
> $
>
> where $\hat{W}$ contains unit-norm columns and $M \in \mathbb{R}^{1 \times C_{\text{out}}}$ stores the magnitudes. During adaptation, we freeze $\hat{W}$ and update $M$ along with a learnable orthogonal matrix $O \in \mathbb{R}^{C_{\text{out}} \times C_{\text{out}}}$, constructed via Householder reflections. The updated weights become:
>
> $
> W' = M \odot (\hat{W} \cdot O).
> $
>
> Here, $O$ operates along the output channel dimension. The spatial structure within each filter remains intact because we do not rotate across spatial positions. Notably, flattening convolution filters is a standard practice in low-rank adaptation and model compression literature. Reshaping back ensures that spatial locality is fully preserved during convolution.
>
> To **validate** this extension, we follow standard CTTA setups used in prior works: WideResNet-28 on CIFAR-10-C, ResNeXt-29 on CIFAR-100-C, and ResNet-50 on ImageNet-C. We select representative and open-sourced baselines for comparison, using benchmark results that have been consistently reported and validated across multiple CTTA studies. The results show **consistent performance gains** across all convolutional backbones, indicating that PAID generalizes beyond ViTs and is also applicable to CNNs.
>
> | Dataset     | Source | TENT[31]  | CoTTA[9] | AdaCon[36] | CRG(2) | LAW(3) | Ours |
> |-------------|--------|-------|-------|--------|--------|------|------|
> | ImageNet-C  | 82.0   | 62.6  | 62.7  | 65.5   | 59.1   | 60.1   | **58.4** |
> | CIFAR100-C  | 46.4   | 60.9  | 32.5  | 33.4   | 29.0   | 30.9   | **28.1** |
> | CIFAR10-C   | 43.5   | 20.7  | 16.1  | 18.5   | 15.9   | 15.7   | **15.0** |
>
> To further **explain** why PAID remains valid in convolutional networks, we revisit several prior studies on the geometric structure of convolutional filters. DCNet [18] shows that the convolution operation can be reformulated as an inner product and decomposed into magnitude and angular: the magnitude better models intra-class variation, while the angular captures semantic differences. SphereConv [15] further demonstrates that learning only the angular component of convolutional filters is sufficient for semantic classification. These findings indicate that the angular structure in CNNs also carries domain-invariant semantics. Therefore, applying PAID’s strategy to convolutional networks is not only formally valid, but also **empirically supported** by prior work.
>
>
> ## **Q4: Limited performance improvement in Table 1 and Table 4**
>
> We sincerely thank the reviewer for the rigorous examination of the results. In response to the concern, we would like to clarify its significance from two perspectives: **empirical consistency** and **research value**.
>
> ### First, the performance improvements are consistent across datasets and tasks, rather than accidental fluctuations.
>
> Our method achieves a +1.5% gain on the CIFAR100-C benchmark, and a +1.6% gain on CIFAR10-C. On the ImageNet-C benchmark, which is the largest and most diverse, our method maintains a 0.3% lower mean error than C-MAE. Moreover, on the ACDC segmentation benchmark, our method outperforms ViDA by +0.3% in mIoU. These **consistent improvements across different datasets and tasks** demonstrate the **robustness** and **generalizability** of our method, suggesting that its effectiveness does not rely on any data distribution from a particular benchmark.
>
> ### Second, the core contribution of this paper lies in proposing a novel direction for CTTA research, which we believe carries substantial value.
>
> While most existing methods focus on exploiting target stream data, PAID is the **first** to explore **pre-trained weights** as more than static initializations. By preserving the pairwise angular structure of weight directions during adaptation, PAID enables a structured update that respects the semantic organization encoded during pre-training. This perspective **establishes a new design paradigm** for CTTA, with the potential to **inspire** a range of principled adaptation strategies and lay a solid foundation for future performance improvements.
>
> (1) On Pitfalls of Test-time Adaptation, ICLR 2023.
>
> (2) Reshaping the Online Data Buffering and Organizing Mechanism for Continual Test-Time Adaptation, ECCV 2024.
>
> (3) Layer-wise Auto-Weighting for Non-Stationary Test-Time Adaptation, WACV 2024.

---

> > ### Comment · Reviewer_jqAQ · 2025-08-05
> >
> > Thanks to the authors for providing a thorough response that addresses all of my concerns. I would therefore like to raise my final rating.

---

> > > ### Author Response · Authors · 2025-08-05
> > > **Thank you for raising the final score.**
> > >
> > > We sincerely thank you for raising the final score and for taking the time to engage with our rebuttal. From your review, we could clearly see your deep understanding of the CTTA field and your high level of expertise. We are also very pleased to have had the opportunity to discuss our work with you.
> > >
> > > You mainly raised four points: the performance degradation of CTTA methods under small batch sizes, the sensitivity of CTTA methods to hyperparameters, the generalizability of our approach to convolutional networks, and the discussion on performance gains. Through our exchanges, we have learned a great deal and have also further clarified several important research questions for the CTTA community. These include strategies for handling small batch sizes and automatic hyperparameter tuning in real-world scenarios. In the revised manuscript, we will incorporate all the additional experiments and detailed analyses that you suggested.
> > >
> > > It is our sincere hope that this work can introduce a new design paradigm and provide inspiration for the CTTA community. Unlike most existing methods that focus primarily on exploiting target stream data, our work is the first to treat pre-trained weights as more than static initializations. This enables a structured update process that preserves the semantic organization encoded during pre-training. We also believe that the insights presented here have the potential to extend to other areas, such as transfer learning and knowledge retention, thus contributing more broadly to the community.
> > >
> > > We would be very glad to discuss any further aspects you might wish to explore. Your thoughtful and constructive feedback has been invaluable, and we look forward to continuing to contribute to the advancement of research within the CTTA community.
> > >
> > > Thank you once again for your constructive engagement throughout this process.

---

> ### Comment · Area_Chair_jJVz · 2025-08-05
> **Reminder to Respond to the Rebuttal**
>
> Dear Reviewer jqAQ,
>
> The discussion phase ends in ~48 hours and you have not engaged with the rebuttal. It is a necessary and important part of the review process and the responsibility of a reviewer for NeurIPS 2025.
>
> Please remember to acknowledge the rebuttal and update your review by the discussion deadline. I look forward to your updated review!
>
> **Recall that the rebuttal acknowledgement, final rating, and final justification are required by NeurIPS 2025**.
>
> Best,
> Your AC

---

### Official Review · Reviewer_KZqM · 2025-07-07

**Clarity:** 2
**Significance:** 3
**Originality:** 3
**Rating:** 4
**Confidence:** 4

**Summary:**

This paper tackles the problem of Continual Test-Time Adaptation (CTTA) for vision models. Building on an empirical finding that pairwise angular relationships among weight vectors remain stable across diverse corruption domains while changing primarily with semantic shifts, the authors propose PAID (Pairwise Angular-Invariant Decomposition). PAID factorizes each linear weight into a magnitude term and a direction term, then freezes the original pairwise angles and adapts only through a global orthogonal rotation—implemented via chains of Householder reflections—plus magnitude updates. This design preserves the geometry believed to encode domain-invariant semantics while allowing flexibility to absorb distribution drift. Extensive experiments on ImageNet-C, CIFAR10/100-C, and Cityscapes demonstrate the effectiveness of the method.

**Questions:**

Please refer to the Weaknesses.

**Ethical Concerns:**

["NO or VERY MINOR ethics concerns only"]

**Final Justification:**

The authors have adequately addressed my concerns by providing additional experiments using the ResNet backbone, as well as offering further clarification of their proposed approach. I will maintain my score of acceptance.

**Quality:**

3

**Strengths And Weaknesses:**

Positive Points:
1. The work approaches CTTA from the weight-space geometry perspective, identifying pairwise angles as a domain-invariant signal and building an algorithm tightly linked to that observation.
2. Extensive experiments demonstrate the effectiveness of the proposed method.

Negative Points:
1. The details regarding the feature alignment process described in Section 3.3 require further clarification. Specifically, it is unclear whether Z_t^T denotes features extracted from layer T for sample t within a batch. Additionally, it would be helpful to specify whether source domain features are extracted in the same manner (i.e., from the same layer) or if only the final layer features are used.
2. The authors mainly conduct experiments on the Vit-Based model, does this method can be applied on ResNet models on the ImageNet-C, Cifar-100-C and Cifar-10-C datasets?
3. The authors employ Householder reflection to construct the orthogonal transformation matrix. It would be beneficial to clarify the motivation behind choosing this particular formulation. Are there specific advantages it offers over alternative methods for generating orthogonal matrices? Furthermore, it would be useful to analyze whether the performance of the method is sensitive to the choice of orthogonal transformation strategy.

---

> ### Author Rebuttal · Authors · 2025-07-31
>
> Thank you for your valuable and constructive comments. We appreciate your recognition of our work's novel perspective on CTTA from the view of weight-space geometry and the tight algorithmic design based on this insight. Below, we address your concerns and questions individually:
>
> ## **Q1: Clarity about the $Z_{t}^{T}$ in Eq. (9) and the procedure for computing source and target statistics**
> Thank you very much for pointing out this ambiguity. We agree that our notation in Eq. (9) was unclear and could lead to confusion.
>
> To clarify: in $Z_{t}^{T}$, the subscript \(t\) indicates that the features are from the target domain (in contrast to source domain \(s\)), and the superscript \(T\) refers to the test-time step, meaning the index of the currently arriving target batch in the continual adaptation process.
>
> For each incoming batch $B_{t}^{T}$, we extract features using the **CLS token output after the final layer normalization and before the classification head** in the ViT-Base. This CLS token serves as the feature representation for each image in the batch.
>
> The source domain statistics $(\mu_s, \sigma_s)$ are computed in exactly the same way: we randomly sample 500 source images prior to test-time, extract their CLS token features from the same location, and compute their mean and standard deviation. These statistics are pre-computed once and reused throughout adaptation, with no further access to the source data.
>
> We will revise Section 3.3 to explicitly clarify these. We will also release our code on GitHub so that the details can be better understood.
>
> ## **Q2: Extension to convolutional networks**
> Thank you for your forward-looking suggestion. The key insight of PAID, preserving the pairwise angular structure between weight vectors to retain domain-invariant semantics, can be **formally** extended to convolutional networks, where each convolutional layer is also a parameterized linear transformation.
>
> In a standard convolutional layer, each output channel corresponds to a filter of shape $(C_{\text{in}}, k_h, k_w)$. We flatten this into a vector of length $D = C_{\text{in}} \cdot k_h \cdot k_w$, treating each filter as a vector in $\mathbb{R}^D$. Stacking these column-wise yields a weight matrix $W \in \mathbb{R}^{D \times C_{\text{out}}}$, which we decompose as:
>
> $
> W = M \odot \hat{W},
> $
>
> where $\hat{W}$ contains unit-norm columns and $M \in \mathbb{R}^{1 \times C_{\text{out}}}$ stores the magnitudes. During adaptation, we freeze $\hat{W}$ and update $M$ along with a learnable orthogonal matrix $O \in \mathbb{R}^{C_{\text{out}} \times C_{\text{out}}}$, constructed via Householder reflections. The updated weights become:
>
> $
> W' = M \odot (\hat{W} \cdot O).
> $
>
> Here, $O$ operates along the output channel dimension. The spatial structure within each filter remains intact because we do not rotate across spatial positions.
> Notably, flattening convolution filters and treating them as vectors is a standard practice in low-rank adaptation and model compression literature. Reshaping back after rotation ensures that spatial locality is fully preserved during convolution.
>
> To **validate** this extension, we follow standard CTTA setups used in prior works: WideResNet-28 on CIFAR-10-C, ResNeXt-29 on CIFAR-100-C, and ResNet-50 on ImageNet-C. We select representative and open-sourced baselines for comparison, using benchmark results that have been consistently reported and validated across multiple CTTA studies. The results show **consistent performance gains** across all convolutional backbones, indicating that PAID generalizes beyond ViTs and is also applicable to CNNs.
>
> | Dataset     | Model          | Source | TENT[31]  | CoTTA[9] | AdaCon[36] | CRG(1) | LAW(2) | Ours |
> |-------------|----------------|--------|-------|-------|--------|--------|------|------|
> | ImageNet-C  | ResNet-50       | 82.0   | 62.6  | 62.7  | 65.5   | 59.1   | 60.1   | **58.4** |
> | CIFAR100-C   | ResNeXt-29    | 46.4   | 60.9  | 32.5  | 33.4   | 29.0   | 30.9   | **28.1** |
> | CIFAR10-C  | WideResNet-28   | 43.5   | 20.7  | 16.1  | 18.5   | 15.9   | 15.7   | **15.0** |
>
> To further **explain** why PAID remains valid in convolutional networks, we revisit several prior studies on the geometric structure of convolutional filters. DCNet [18] shows that the convolution operation can be reformulated as an inner product and decomposed into magnitude and angular: the magnitude better models intra-class variation, while the angular captures semantic differences. SphereConv [15] further demonstrates that learning only the angular component of convolutional filters is sufficient for semantic classification. These findings indicate that the angular structure in CNNs also carries domain-invariant semantics. Therefore, applying PAID’s strategy to convolutional networks is not only formally valid, but also **empirically supported** by prior work.
>
> ## **Q3: Motivation and sensitivity analysis of using householder reflection for orthogonal transformation**
>
> We thank the reviewer for this constructive question. We choose to use the Householder reflection primarily due to **computational efficiency** considerations. Below, we elaborate on our motivation through a comparison with alternative implementations and an analysis of sensitivity.
>
> ### Alternative: cayley transformation
> Another common mathmatics tool for constructing orthogonal matrices $ R \in \mathbb{R}^{d \times d} $ is the cayley transformation, which maps a skew-symmetric matrix $ A \in \mathbb{R}^{d \times d} $, where $ A^\top = -A $, to an orthogonal matrix:
>
> $
> R = (I + A)(I - A)^{-1}.
> $
>
> This formulation underlies two recently proposed methods:
> - **OFT** [20]: Constructs an orthogonal matrix by applying the Cayley transformation to multiple learnable skew-symmetric blocks.
> - **BOFT** (3): Builds a butterfly-style composition of sparse Cayley-transformed blocks for efficient approximation.
>
> ### Comparison of parameter and computational complexity
> We refer to the analysis in Section 3.2 and Table 1 of [21], which compares these two types of methods:
> - **Parameter complexity**: Both Cayley-based methods and the Householder-based method require approximately $ \mathcal{O}(d^2) $ parameters.
> - **Computational complexity**: Cayley-based methods involve matrix inversions and dense multiplications, resulting in **$ \mathcal{O}(d^3) $** complexity. In contrast, the Householder-based method only involves vector inner products and scale-vector multiplications, yielding a lower cost of **$ \mathcal{O}(d^2) $**.
>
> This **one-order-of-magnitude improvement** in computational complexity makes Householder reflection a more efficient choice for constructing orthogonal matrices.
>
> ### Sensitivity analysis
> We construct orthogonal matrices following [20] and (3), and conduct a comparative experiment on the ImageNet-C benchmark. All methods perform similarly, with only a slight performance drop, suggesting that the overall performance is **not sensitive** to the specific choice of orthogonalization strategy.
>
> | Method        | Type         | Mean Error on ImageNet-C  |
> |---------------|--------------|-----------------------------|
> | OFT [20]       | Cayley      |             42.9            |
> | BOFT (3)      | Cayley       |             42.4            |
> | Ours          | Householder  |             42.2            |
>
> That said, we recommend the Householder reflection approach. Given its **comparable expressiveness and parameter complexity** to Cayley-based methods, its **lower computational cost** makes it more suitable for CTTA, where efficient online adaptation and reduced overhead are essential.
>
> (1) Reshaping the Online Data Buffering and Organizing Mechanism for Continual Test-Time
> Adaptation, ECCV 2024.
>
> (2) Layer-wise Auto-Weighting for Non-Stationary Test-Time Adaptation, WACV 2024.
>
> (3) Parameter-efficient Orthogonal Finetuning via Butterfly Factorization, ICLR 2024.

---

> > ### Comment · Reviewer_KZqM · 2025-08-04
> > **Response to Rebuttal**
> >
> > I appreciate the authors’ detailed responses. They have adequately addressed my concern, and I will retain my positive score.

---

> > > ### Author Response · Authors · 2025-08-05
> > >
> > > Thank you very much for your positive follow-up and for taking the time to engage with our rebuttal. You have raised three important points: clarifying specific details of the paper, extending the method to other backbones, and discussing the sensitivity of the orthogonal matrix construction. We are very pleased to hear that our response has addressed your concerns. We fully agree that these comments are essential for enhancing the clarity and persuasiveness of the paper, as well as for broadening its applicability. In the revised manuscript, we will incorporate all the additional experiments and detailed analyses that you suggested.
> > >
> > > It is our sincere hope that this work can introduce a new design paradigm and inspiration for the CTTA community. In contrast to most existing methods that focus on exploiting target stream data, our work is the first to treat pre-trained weights as more than static initializations. This enables a structured update process that preserves the semantic organization encoded during pre-training. We also believe that the insights presented here have the potential to be extended to other areas, such as transfer learning and knowledge retention, thereby contributing more broadly to the community.
> > >
> > > We would be delighted to discuss any further aspects you may wish to explore. Your thoughtful and constructive feedback has been invaluable, and we look forward to contributing further to the advancement of research within the CTTA community.
> > >
> > > Thank you once again for your constructive engagement throughout this process.

---

### Note · Authors · 2025-08-12

We would like to sincerely thank **Area Chair jJVz** and **Reviewers KZqM, jqAQ, e8tG, and tGtE** for their highly constructive suggestions during the review process and for their active engagement during the rebuttal stage. We are also very pleased that all of your questions were fully addressed by the end of the rebuttal, and that we received both score increases and **all-positive** scores. Your constructive feedback has further improved the quality of this work, and we will incorporate the improvements based on your suggestions into the revised version.

The core insight of this paper is that the pairwise angular structure in pre-trained weights encodes domain-invariant semantic prior that should be preserved during continual adaptation to diverse corrupted domains. This preservation prevents the drift of key semantic capabilities in the model during continual adaptation, thereby mitigating catastrophic forgetting and error accumulation. We validate this insight through **theoretical justification, statistical analysis, functional validation, visual interpretation, and intuitive explanation**.

From an experimental perspective, the proposed insight generalizes to and is applicable for **both CNN-based and Transformer-based architectures**, achieving SOTA results on four widely used benchmarks.

It is our sincere hope that this work will **introduce a new design paradigm and inspire the CTTA community**. Unlike most existing methods that primarily focus on exploiting target-stream data, our work is the **first** to regard pre-trained weights as more than static initializations, enabling a structured update process that preserves the semantic organization encoded during pre-training. We also believe the presented insights can **potentially extend to other areas** such as transfer learning and knowledge retention, thereby contributing more broadly to the community.

Thank you once again for your constructive engagement throughout this process.

---

### Decision · Program_Chairs · 2025-09-17

**Decision:**

Accept (poster)

**Comment:**

Pairwise Angular-Invariant Decomposition (PAID) examines the source model parameters when applied to continual adaptation problems for target data and contributes a new prior / regularizer for test-time adaptation methods. The choice of prior—pairwise angular structure—is empirically explored and well motivated across shifts with clear notations and visualizations to support this choice and check alternatives (the magnitude or absolute angle). The experiments show improvement on standard benchmarks for continual test-time adaptation (ImageNet-C, CIFAR-100-C for classification and Cityscapes/ACDC for segmentation) with current models (ViTs, Segformer).

- Positives: the parameterization focus in concept/experiment/theory is complementary to existing work on test-time adaptation (KZqM, jqAQ, e8tG, tGtE), the scenario is practical and the experiments extensive (KZqM, jqAQ), the exposition is clear (jqAQ, e8tG, tGtE).
- Negatives: the work focuses on transformers like ViTs which are current but is not as comprehensive as it could be given the number of existing results for ResNets which are relevant for their efficiency (KZqM, jqAQ, e8tG, tGtE), the results suffer as batch size decreases and the method only addresses batch-wise and not input-wise adaptation (jqAQ), insufficient clarity in the presentation of the method (KZqM), sensitivity to hyperparameters and the need for tuning (jqAQ), and limited theoretical support (e8tG) although this is common for the topic.

The reviewers all converge to ratings of 4: borderline accept and their are broad points of agreement on the positives and the negatives. In particular all reviewers agree on the positive that the parameter geometry perspective is informative and complementary to existing work on continual test-time adaptation and all reviewers agree on the negative that the paper narrows its evaluation too much w.r.t. model architecture. The rebuttal and discussion phases confirmed and clarified the core contribution of PAID on parameter geometry analysis and regularization and resolved the negative about ViT exclusivity with multiple results on convolutional architectures. Reviewers KZqM, jqAQ, e8tG, and tGtE all confirm that the rebuttal has addressed their points and either raise or maintain their score. The area chair sides with acceptance due to the universal positive ratings and agreement on the complementary and informativeness of the main contribution about parameter geometry and variation vs. invariance across shifts. The potential arguments for rejection, like the narrow selection of models for experiment or points of clarity, have all been resolved by the rebuttal as confirmed by the reviewers. The additional results and details should be incorporated into the revision and related work like DoRA and OFT (jqAQ) should be cited as similar in technique if not in purpose and experimental scope.